# GENERATIVE MODELS OF VISUALLY GROUNDED IMAGINATION

**Ramakrishna Vedantam**[*]
Georgia Tech
vrama@gatech.edu

**Ian Fischer**
Google Inc.
iansf@google.com

**Jonathan Huang**
Google Inc.
jonathanhuang@google.com

**Kevin Murphy**
Google Inc.
kpmurphy@google.com

## ABSTRACT

It is easy for people to imagine what a man with pink hair looks like, even if they have never seen such a person before. We call the ability to create images of novel semantic concepts *visually grounded imagination*. In this paper, we show how we can modify variational auto-encoders to perform this task. Our method uses a novel training objective, and a novel product-of-experts inference network, which can handle partially specified (abstract) concepts in a principled and efficient way. We also propose a set of easy-to-compute evaluation metrics that capture our intuitive notions of what it means to have good visual imagination, namely correctness, coverage, and compositionality (the *3 C's*). Finally, we perform a detailed comparison of our method with two existing joint image-attribute VAE methods (the JMVAE method of Suzuki et al. (2017) and the BiVCCA method of Wang et al. (2016b)) by applying them to two datasets: the MNIST-with-attributes dataset (which we introduce here), and the CelebA dataset (Liu et al., 2015).

## 1 INTRODUCTION

Consider the following two-party communication game: a speaker thinks of a visual concept $C$, such as "men with black hair", and then generates a description $\mathbf{y}$ of this concept, which she sends to a listener; the listener interprets the description $\mathbf{y}$, by creating an internal representation $\mathbf{z}$, which captures its "meaning". We can think of $\mathbf{z}$ as representing a set of "mental images" which depict the concept $C$. To test whether the listener has correctly "understood" the concept, we ask him to draw a set of real images $\mathcal{S} = \{\mathbf{x}_s : s = 1 : S\}$, which depict the concept $C$. He then sends these back to the speaker, who checks to see if the images correctly match the concept $C$. We call this process *visually grounded imagination*.

In this paper, we represent concept descriptions in terms of a fixed length vector of discrete attributes $\mathcal{A}$. This allows us to specify an exponentially large set of concepts using a compact, combinatorial representation. In particular, by specifying different subsets of attributes, we can generate concepts at different levels of granularity or abstraction. We can arrange these concepts into a *compositional abstraction hierarchy*, as shown in Figure 1. This is a directed acyclic graph (DAG) in which nodes represent concepts, and an edge from a node to its parent is added whenever we drop one of the attributes from the child's concept definition. Note that we dont make any assumptions about the order in which the attributes are dropped (that is, dropping the attribute "smiling" is just as valid as dropping "female" in Figure 1). Thus, the tree shown in the figure is just a subset extracted from the full DAG of concepts, shown for illustration purposes.

We can describe a concept by creating the attribute vector $\mathbf{y}_{\mathcal{O}}$, in which we only specify the value of the attributes in the subset $\mathcal{O} \subseteq \mathcal{A}$; the remaining attributes are unspecified, and are assumed to take all possible legal values. For example, consider the following concepts, in order of increasing abstraction: $C_{msb} = (\text{male}, \text{smiling}, \text{blackhair})$, $C_{*sb} = (*, \text{smiling}, \text{blackhair})$, and $C_{**b} = (*, *, \text{blackhair})$,

---

[*]Work performed during an internship at Google.

where the attributes are gender, smiling or not, and hair color, and $*$ represents "don't care". A good model should be able to generate images from different levels of the abstraction hierarchy, as shown in Figure 1. (This is in contrast to most prior work on conditional generative models of images, which assume that all attributes are fully specified, which corresponds to sampling only from leaf nodes in the hierarchy.)

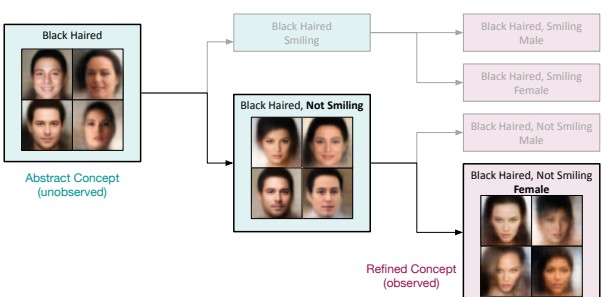

Figure 1: A compositional abstraction hierarchy for faces, derived from 3 attributes: hair color, smiling or not, and gender. We show a set of sample images generated by our model, when trained on CelebA, for different nodes in this hierarchy.

In Section 2, we show how we can extend the variational autoencoder (VAE) framework of Kingma & Welling (2014) to create models which can perform this task. The first extension is to modify the model to the "multi-modal" setting where we have both an image, $\mathbf{x}$, and an attribute vector, $\mathbf{y}$. More precisely, we assume a joint generative model of the form $p(\mathbf{x}, \mathbf{y}, \mathbf{z}) = p(\mathbf{z})p(\mathbf{x}|\mathbf{z})p(\mathbf{y}|\mathbf{z})$, where $p(\mathbf{z})$ is the prior over the latent variable $\mathbf{z}$, $p(\mathbf{x}|\mathbf{z})$ is our image decoder, and $p(\mathbf{y}|\mathbf{z})$ is our description decoder. We additionally assume that the description decoder factorizes over the specified attributes in the description, so $p(\mathbf{y}_{\mathcal{O}}|\mathbf{z}) = \prod_{k \in \mathcal{O}} p(y_k|\mathbf{z})$.

We further extend the VAE by devising a novel objective function, which we call the *TELBO*, for training the model from paired data, $\mathcal{D} = \{(\mathbf{x}_n, \mathbf{y}_n)\}$. However, at test time, we will allow unpaired data (either just a description or just an image). Hence we fit three inference networks: $q(\mathbf{z}|\mathbf{x}, \mathbf{y})$, $q(\mathbf{z}|\mathbf{x})$ and $q(\mathbf{z}|\mathbf{y})$. This way we can embed an image or a description into the same shared latent space (using $q(\mathbf{z}|\mathbf{x})$ and $q(\mathbf{z}|\mathbf{y})$, respectively); this lets us "translate" images into descriptions or vice versa, by computing $p(\mathbf{y}|\mathbf{x}) = \int d\mathbf{z} \, p(\mathbf{y}|\mathbf{z})q(\mathbf{z}|\mathbf{x})$ and $p(\mathbf{x}|\mathbf{y}) = \int d\mathbf{z} \, p(\mathbf{x}|\mathbf{z})q(\mathbf{z}|\mathbf{y})$.

To handle abstract concepts (i.e., partially observed attribute vectors), we use a method based on the product of experts (POE) (Hinton, 2002). In particular, our inference network for attributes has the form $q(\mathbf{z}|\mathbf{y}_{\mathcal{O}}) \propto p(\mathbf{z}) \prod_{k \in \mathcal{O}} q(\mathbf{z}|\mathbf{y}_k)$. If no attributes are specified, the posterior is equal to the prior. As we condition on more attributes, the posterior becomes narrower, which corresponds to specifying a more precise concept. This enables us to generate a more diverse set of images to represent abstract concepts, and a less diverse set of images to represent concrete concepts, as we show below.

Section 3 discusses how to evaluate the performance of our method in an objective way. Specifically, we first "ground" the description by generating a set of images, $\mathcal{S}(\mathbf{y}_{\mathcal{O}}) = \{\mathbf{x}^s \sim p(\mathbf{x}|\mathbf{y}_{\mathcal{O}}) : s = 1 : S\}$. We then check that all the sampled images in $\mathcal{S}(\mathbf{y}_{\mathcal{O}})$ are consistent with the specified attributes $\mathbf{y}_{\mathcal{O}}$ (we call this **correctness**). We also check that the set of images "spans" the extension of the concept, by exhibiting suitable diversity (c.f. (Young et al., 2014)). Concretely, we check that the attributes that were *not specified* (e.g., gender in $C_{*sb}$ above) vary across the different images; we call this **coverage**. Finally, we want the set of images to have high correctness and coverage even if the concept $\mathbf{y}_{\mathcal{O}}$ has a combination of attribute values that have not been seen in training. For example, if we train on $C_{msb} = (\text{male}, \text{smiling}, \text{blackhair})$, and $C_{fnb} = (\text{female}, \text{notsmiling}, \text{blackhair})$, we should be able to test on $C_{mnb} = (\text{male}, \text{notsmiling}, \text{blackhair})$, and $C_{fsb} = (\text{female}, \text{smiling}, \text{blackhair})$. We will call this property **compositionality**. Being able to generate plausible images in response to truly compositionally novel queries is the essence of imagination. Together, we call these criteria *the 3 C's of visual imagination*.

Section 5 reports experimental results on two different datasets. The first dataset is a modified version of MNIST, which we call MNIST-with-attributes (or MNIST-A), in which we "render" modified versions of a single MNIST digit on a 64x64 canvas, varying its location, orientation and size. The

second dataset is CelebA (Liu et al., 2015), which consists of over 200k face images, annotated with 40 binary attributes. We show that our method outperforms previous methods on these datasets.

The contributions of this paper are threefold. First, we present a novel extension to VAEs in the multimodal setting, introducing a principled new training objective (the TELBO), and deriving an interpretation of a previously proposed objective (JMVAE) (Wang et al., 2016a) as a valid alternative in Appendix A.1. Second, we present a novel way to handle missing data in inference networks based on a product of experts. Third, we present novel criteria (the 3 C's) for evaluating conditional generative models of images, that extends prior work by considering the notion of visual abstraction and imagination.

## 2 METHODS

We start by describing standard VAEs, to introduce notation. We then discuss our extensions to handle the multimodal and the missing input settings.

**Standard VAEs.** A variational autoencoder (Kingma & Welling, 2014) is a latent variable model of the form $p_{\boldsymbol{\theta}}(\mathbf{x}, \mathbf{z}) = p_{\boldsymbol{\theta}}(\mathbf{z})p_{\boldsymbol{\theta}}(\mathbf{x}|\mathbf{z})$, where $p_{\boldsymbol{\theta}}(\mathbf{z})$ is the prior (we assume it is Gaussian, $p_{\boldsymbol{\theta}}(\mathbf{z}) = \mathcal{N}(\mathbf{z}|\mathbf{0}, \mathbf{I})$, although this assumption can be relaxed), and $p_{\boldsymbol{\theta}}(\mathbf{x}|\mathbf{z})$ is the likelihood (sometimes called the decoder), usually represented by a neural network. To perform approximate posterior inference, we fit an inference network (sometimes called the encoder) of the form $q_{\boldsymbol{\phi}}(\mathbf{z}|\mathbf{x})$, so as to maximize $\mathcal{L}(\boldsymbol{\theta}, \boldsymbol{\phi}) = \mathbb{E}_{\hat{p}(\mathbf{x})}[\mathrm{elbo}(\mathbf{x}, \boldsymbol{\theta}, \boldsymbol{\phi})]$, where $\hat{p}(\mathbf{x}) = \frac{1}{N}\sum_{n=1}^{N}\delta_{\mathbf{x}_n}(\mathbf{x})$ is the empirical distribution, and ELBO is the evidence lower bound:

$$\mathrm{elbo}_{\lambda,\beta}(\mathbf{x}, \boldsymbol{\theta}, \boldsymbol{\phi}) = \mathbb{E}_{q_{\boldsymbol{\phi}}(\mathbf{z}|\mathbf{x},\boldsymbol{\phi})}[\lambda \log p_{\boldsymbol{\theta}}(\mathbf{x}|\mathbf{z})] - \beta\mathrm{KL}(q_{\boldsymbol{\phi}}(\mathbf{z}|\mathbf{x}), p_{\boldsymbol{\theta}}(\mathbf{z})) \qquad (1)$$

Here $\mathrm{KL}(p, q)$ is the Kullback Leibler divergence between distributions $p$ and $q$. By default, $\beta = \lambda = 1$, in which case we will just write $\mathrm{elbo}(\mathbf{x}, \boldsymbol{\theta}, \boldsymbol{\phi})$. However, by using $\beta > 1$ we can encourage the posterior to be closer to the factorial prior $p(\mathbf{z}) = \mathcal{N}(\mathbf{z}|\mathbf{0}, \mathbf{I})$, which encouarges the latent factors to be "disentangled", as proved in Achille & Soatto (2017); this is known as the $\beta$-VAE trick (Higgins et al., 2017a). And allowing $\lambda > 1$ will be useful later, when we have multiple modalities.

**Joint VAEs and the TELBO.** We extend the VAE to model images and attributes by defining the joint distribution $p_{\boldsymbol{\theta}}(\mathbf{x}, \mathbf{y}, \mathbf{z}) = p_{\boldsymbol{\theta}}(\mathbf{z})p_{\boldsymbol{\theta}}(\mathbf{x}|\mathbf{z})p_{\boldsymbol{\theta}}(\mathbf{y}|\mathbf{z})$, where $p_{\boldsymbol{\theta}}(\mathbf{x}|\mathbf{z})$ is the image decoder (we use the DCGAN architecture from Radford et al. (2016)), and $p_{\boldsymbol{\theta}}(\mathbf{y}|\mathbf{z})$ is an MLP for the attribute vector. The corresponding training objective which we want to maximize becomes $\mathcal{L}(\boldsymbol{\theta}, \boldsymbol{\phi}) = \mathbb{E}_{\hat{p}(\mathbf{x},\mathbf{y})}[\mathrm{elbo}(\mathbf{x}, \mathbf{y}, \boldsymbol{\theta}, \boldsymbol{\phi})]$, where $\hat{p}(\mathbf{x}, \mathbf{y}) = \frac{1}{N}\sum_{n=1}^{N}\delta_{\mathbf{x}_n}(\mathbf{x})\delta_{\mathbf{y}_n}(\mathbf{y})$ is the empirical distribution derived from paired data, and the joint ELBO is given by

$$\mathrm{elbo}_{\lambda_x,\lambda_y,\beta}(\mathbf{x}, \mathbf{y}, \boldsymbol{\theta}_x, \boldsymbol{\theta}_y, \boldsymbol{\phi}) = \mathbb{E}_{q_{\boldsymbol{\phi}}(\mathbf{z}|\mathbf{x},\mathbf{y})}\left[\lambda_x \log p_{\boldsymbol{\theta}_x}(\mathbf{x}|\mathbf{z}) + \lambda_y \log p_{\boldsymbol{\theta}_y}(\mathbf{y}|\mathbf{z})\right]$$
$$- \beta\mathrm{KL}(q_{\boldsymbol{\phi}}(\mathbf{z}|\mathbf{x}, \mathbf{y}), p_{\boldsymbol{\theta}}(\mathbf{z}))$$

We call this the JVAE (joint VAE) model. We usually set $\beta = 1$, but set $\lambda_y/\lambda_x > 1$ to to scale up the likelihood from the low dimensional attribute vector, $p_{\boldsymbol{\theta}}(\mathbf{y}|\mathbf{z})$, to match the likelihood from the high dimensional image, $p_{\boldsymbol{\theta}}(\mathbf{x}|\mathbf{z})$.

Having fit the joint model above, we can proceed to train unpaired inference networks $q_{\boldsymbol{\phi}_x}(\mathbf{z}|\mathbf{x})$ and $q_{\boldsymbol{\phi}_y}(\mathbf{z}|\mathbf{y})$, so we can embed images and attributes into the same shared latent space. Keeping the $p$ family fixed from the joint model, a natural objective to fit, say, $q_{\boldsymbol{\phi}_x}(\mathbf{z}|\mathbf{x})$ is to maximize the following:[1]

$$\mathcal{L}(\boldsymbol{\phi}_x|\boldsymbol{\theta}) = -\mathbb{E}_{\hat{p}(\mathbf{x})}\left[\mathrm{KL}(q_{\boldsymbol{\phi}_x}(\mathbf{z}|\mathbf{x}), p_{\boldsymbol{\theta}_x}(\mathbf{z}|\mathbf{x}))\right]$$
$$= \int\int d\mathbf{x}d\mathbf{z}\ \hat{p}(\mathbf{x})q_{\boldsymbol{\phi}_x}(\mathbf{z}|\mathbf{x})\left[-\log q_{\boldsymbol{\phi}_x}(\mathbf{z}|\mathbf{x}) - \log p_{\boldsymbol{\theta}_x}(\mathbf{x}) + \log p_{\boldsymbol{\theta}_x}(\mathbf{x}|\mathbf{z}) + \log p_{\boldsymbol{\theta}}(\mathbf{z})\right]$$
$$= \mathbb{E}_{\hat{p}(\mathbf{x})}\left[\mathrm{elbo}(\mathbf{x}, \boldsymbol{\theta}_x, \boldsymbol{\phi}_x)\right] - \mathbb{E}_{\hat{p}(\mathbf{x})}\left[\log p_{\boldsymbol{\theta}_x}(\mathbf{x})\right]$$

---

[1] A reasonable alternative would be to minimize $\mathbb{E}_{\hat{p}(\mathbf{x})}[\mathrm{KL}(p_{\boldsymbol{\theta}_x}(\mathbf{z}|\mathbf{x}), q_{\boldsymbol{\phi}_x}(\mathbf{z}|\mathbf{x}))]$. However, this is intractable, since we cannot compute $p_{\boldsymbol{\theta}_x}(\mathbf{z}|\mathbf{x})$, by assumption.

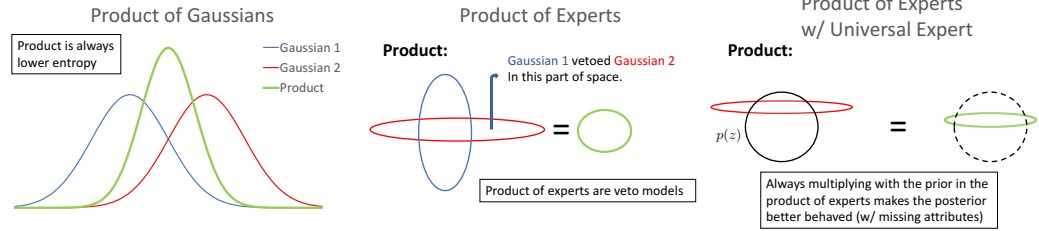

Figure 2: Illustration of the product of experts inference network. Each expert votes for a part of latent space implied by its observed attribute. The final posterior is the intersection of these regions. When all attributes are observed, the posterior will be a narrowly defined Gaussian, but when some attributes are missing, the posterior will be broader. Right: we illustrate how inclusion of the "universal expert" $p(\mathbf{z})$ in the product ensures that the posterior is always well-conditioned (close to spherical), even when we are missing some attributes.

where the last term is constant wrt $\phi_x$ and the model family $p$, and hence can be dropped. We can use a similar method to fit $q_{\phi_y}(\mathbf{z}|\mathbf{y})$. Combining these gives the following triple ELBO (*TELBO*) objective:

$$\mathcal{L}(\boldsymbol{\theta}_x, \boldsymbol{\theta}_y, \boldsymbol{\phi}, \boldsymbol{\phi}_x, \boldsymbol{\phi}_y) = \mathbf{E}_{\hat{p}(\mathbf{x},\mathbf{y})}\left[\mathrm{elbo}_{1,\lambda,1}(\mathbf{x}, \mathbf{y}, \boldsymbol{\theta}_x, \boldsymbol{\theta}_y, \boldsymbol{\phi})\right.$$
$$\left. + \mathrm{elbo}_{1,1}(\mathbf{x}, \boldsymbol{\theta}_x, \boldsymbol{\phi}_x) + \mathrm{elbo}_{\gamma,1}(\mathbf{y}, \boldsymbol{\theta}_y, \boldsymbol{\phi}_y)\right] \quad (2)$$

where $\lambda$ and $\gamma$ scale the log likelihood terms $\log p(\mathbf{y}|\mathbf{z})$; we set these parameters using a validation set. Since we are training the generative model only on aligned data, and simply retrofitting inference networks, we freeze the $p_{\boldsymbol{\theta}_x}(\mathbf{x}|\mathbf{z})$ and $p_{\boldsymbol{\theta}_y}(\mathbf{y}|\mathbf{z})$ terms when training the last two ELBO terms above, and just optimize $q_{\phi_x}(\mathbf{z}|\mathbf{x})$ and $q_{\phi_y}(\mathbf{z}|\mathbf{y})$ terms. This enables us to optimize all terms in Equation (2) jointly. Alternatively, we can first fit the joint model, and then fit the unimodal inference networks.[2] In Section 4, we compare this to other methods for training joint VAEs that have been proposed in the literature.

**Handling missing attributes.** In order to handle missing attributes at test time, we use a product of experts model, where each attribute instantiates an expert. We are motivated by prior work (Williams & Nash, 2018) which shows that for a linear factor analysis model, the posterior distribution $p(\mathbf{z}|\mathbf{y})$ is a product of $K$-dimensional Gaussians, one for each visible dimension. Since our model is just a nonlinear extension of factor analysis, we choose the form of the approximate posterior of our inference network, $q(\mathbf{z}|\mathbf{y})$, to be a product of Gaussians, one for each visible feature: $q(\mathbf{z}|\mathbf{y}_{\mathcal{O}}) \propto p(\mathbf{z}) \prod_{k \in \mathcal{O}} q(\mathbf{z}|y_k)$, where $q(\mathbf{z}|y_k) = \mathcal{N}(\mathbf{z}|\boldsymbol{\mu}_k(y_k), \mathbf{C}_k(y_k))$ is the $k$th Gaussian "expert", and $p(\mathbf{z}) = \mathcal{N}(\mathbf{z}|\boldsymbol{\mu}_0 = \mathbf{0}, \mathbf{C}_0 = \mathbf{I})$ is the prior. A similar model was concurrently proposed in Bouchacourt et al. (2018) to perform inference for a set of images. Unlike the product of experts model in (Hinton, 2002), our model multiplies Gaussians, not Bernoullis, so the product has a closed form solution namely $q(\mathbf{z}|\mathbf{y}_{\mathcal{O}}) = \mathcal{N}(\mathbf{z}|\boldsymbol{\mu}, \mathbf{C})$, where $\mathbf{C}^{-1} = \sum_k \mathbf{C}_k^{-1}$ and $\boldsymbol{\mu} = \mathbf{C}(\sum_k \mathbf{C}_k^{-1}\boldsymbol{\mu}_k)$, and the sum is over all the observed attributes. Intuitively, $\mathbf{y}$ imposes an increasing number of constraints on $\mathbf{z}$ as more of it is observed, as explained in Williams & Agakov (2002). In our setting, if we do not observe any attributes, the posterior reduces to the prior. As we observe more attributes, the posterior becomes narrower, since the (positive definite) precision matrices, $\mathbf{C}^{-1}$ add up, reflecting the increased specificity of the concept being specified, as illustrated in Figure 2 (middle) (see also Williams & Agakov (2002)). We always include the prior term, $p(\mathbf{z})$, in the product, since without it, the posterior $q_{\phi_y}(\mathbf{z}|\mathbf{y}_{\mathcal{O}})$ may not be well-conditioned when we are missing attributes, as illustrated in Figure 2 (right). For more implementation-level details on the model architectures, see Appendix A.4.

## 3 EVALUATION METRICS: THE 3C'S OF VISUAL IMAGINATION

To evaluate the quality of a set of generated images, $\mathcal{S}(\mathbf{y}_{\mathcal{O}}) = \{\mathbf{x}_s \sim p(\mathbf{x}|\mathbf{y}_{\mathcal{O}}) : s = 1 : S\}$, we apply a multi-label classifier to each image, to convert it to a predicted attribute vector, $\hat{\mathbf{y}}(\mathbf{x})$. This

---

[2] If we have unlabeled image data, we can perform semisupervised learning by optimizing $\mathbb{E}_{\hat{p}(\mathbf{x})}\left[\mathrm{elbo}(\mathbf{x}, \boldsymbol{\theta}_x, \boldsymbol{\phi}_x)\right]$ wrt $\boldsymbol{\theta}_x$ and $\boldsymbol{\phi}_x$, as in Pu et al. (2016).

attribute classifier is trained on a large dataset of images and attributes, and is held constant across all methods that are being evaluated. It plays the role of a human observer. This is similar in spirit to generative adversarial networks (Goodfellow et al., 2014), that declare a generated image to be good enough if a binary classifier cannot distinguish it from a real image. (Both approaches avoid the problems mentioned in Theis et al. (2016) related to evaluating generative image models in terms of their likelihood.) However, the attribute classifier checks not only that the images look realistic, but also that they have the desired attributes.

To quantify this, we define the **correctness** as the fraction of attributes for each generated image that match those specified in the concept's description: $\text{correctness}(\mathcal{S}, \mathbf{y}_{\mathcal{O}}) = \frac{1}{|\mathcal{S}|} \sum_{\mathbf{x} \in \mathcal{S}} \frac{1}{|\mathcal{O}|} \sum_{k \in \mathcal{O}} \mathbb{I}(\hat{y}(\mathbf{x})_k = y_k)$. However, we also want to measure the diversity of values for the *unspecified* or missing attributes, $\mathcal{M} = \mathcal{A} \setminus \mathcal{O}$. We do this by comparing $q_k$, the empirical distribution over values for attribute $k$ induced by the generated set $\mathcal{S}$, to $p_k$, the true distribution for this attribute induced by the training set. We measure the difference between these distributions using the Jensen-Shannon divergence, since it is symmetric and satisfies $0 \leq \text{JS}(p, q) \leq 1$. We then define the **coverage** as follows: $\text{coverage}(S, \mathbf{y}_{\mathcal{O}}) = \frac{1}{|\mathcal{M}|} \sum_{k \in \mathcal{M}} (1 - \text{JS}(p_k, q_k))$. If desired, we can combine correctness and coverage into a single number, by computing the JS divergence between $p_k$ and $q_k$ for all attributes, where, for observed attributes, $p_k$ is a delta function and $q_k$ is the empirical distribution (we call this **JS-overall**). This gives us a convenient way to pick hyperparameters. However, for analysis, we find it helpful to report correctness and coverage separately.

Note that our metric is different from the inception score proposed in Salimans et al. (2016). That is defined as follows: $\text{inception} = \exp\left(\mathbb{E}_{\hat{p}(\mathbf{x})}\left[\text{KL}(p(y|\mathbf{x}), p(y))\right]\right)$, where $y$ is a class label. Expanding the term inside the exponential, we get

$$\sum_{\mathbf{x}} p(\mathbf{x}) \left[\sum_{y} p(y|\mathbf{x}) \log p(y|\mathbf{x})\right] - \sum_{\mathbf{x}} \sum_{y} p(\mathbf{x}, y) \log p(y) = \mathbb{E}_{\hat{p}(\mathbf{x})}\left[-H(y|\mathbf{x})\right] + H(y)$$

A high inception score means that the distribution $p(y|\mathbf{x})$ has low entropy, so the generated images match some class, but that the marginal $p(y)$ has high entropy, so the images are diverse. However, the inception score was created to evaluate unconditional generative models of images, so it does not check if the generated images are consistent with the concept $\mathbf{y}_{\mathcal{O}}$, and the degree of diversity does not vary in response to the level of abstraction of the concept.

Finally, we can assess how well the model understands **compositionality**, by checking correctness of its generated images in response to test concepts $\mathbf{y}_{\mathcal{O}}$ that differ in at least one attribute from the training concepts. We call this a *compositional split* of the data. This is much harder than a standard *iid* split, since we are asking the model to predict the effects of novel combinations of attributes, which it has not seen before (and which might actually be impossible). Note that abstraction is different from compositionality – in abstraction we are asking the model to predict the effects of dropping certain attributes instead of predicting novel combinations of attributes.

## 4 RELATED WORK

In this section, we briefly mention some of the most closely related prior work.

**Conditional models.** Many conditional generative image models of the form $p(\mathbf{y}|\mathbf{x})$ have been proposed recently, where $\mathbf{y}$ can be a class label (e.g., (Radford et al., 2016)), a vector of attributes (e.g., (Yan et al., 2016)), a sentence (e.g., (Reed et al., 2016)), another image (e.g., (Isola et al., 2017)), etc. Such models are usually based on VAEs or GANs. However, we are more interested in learning a shared latent space from either descriptions $\mathbf{y}$ or images $\mathbf{x}$, which means we need to use a joint, symmetric, model.

**Joint models.** Several papers use the same joint VAE model as us, but they differ in how it is trained. In particular, the BiVCCA objective of Wang et al. (2016b) has the form $\mathcal{L}(\boldsymbol{\theta}, \boldsymbol{\phi}) = $

$\mathbb{E}_{\hat{p}(\mathbf{x},\mathbf{y})}\left[J(\mathbf{x},\mathbf{y},\boldsymbol{\theta},\boldsymbol{\phi})\right]$, where

$$J(\mathbf{x},\mathbf{y},\boldsymbol{\theta},\boldsymbol{\phi}) = \mu\left(E_{q_{\boldsymbol{\phi}_x}(\mathbf{z}|\mathbf{x})}[\log p_{\boldsymbol{\theta}_x}(\mathbf{x}|\mathbf{z}) + \lambda\log p_{\boldsymbol{\theta}_y}(\mathbf{y}|\mathbf{z})] - \mathrm{KL}(q_{\boldsymbol{\phi}_x}(\mathbf{z}|\mathbf{x}), p_{\boldsymbol{\theta}}(\mathbf{z}))\right)$$
$$+ (1-\mu)\left(E_{q_{\boldsymbol{\phi}_y}(\mathbf{z}|\mathbf{y})}[\log p_{\boldsymbol{\theta}_x}(\mathbf{x}|\mathbf{z}) + \lambda\log p_{\boldsymbol{\theta}_y}(\mathbf{y}|\mathbf{z})] - \mathrm{KL}(q_{\boldsymbol{\phi}_y}(\mathbf{z}|\mathbf{y}), p_{\boldsymbol{\theta}}(\mathbf{z}))\right)$$

This method results in the model generating the mean image corresponding to each concept, due to the $E_{q_{\boldsymbol{\phi}_y}(\mathbf{z}|\mathbf{y})}\log p_{\boldsymbol{\theta}}(\mathbf{x},\mathbf{y}|\mathbf{z})$ term, which requires that $\mathbf{z}$'s sampled from $q_{\boldsymbol{\phi}_y}(\mathbf{z}|\mathbf{y}_n)$ be good at generating all the different $\mathbf{x}_n$'s which co-occur with $\mathbf{y}_n$. We show this empirically in Section 5. This problem can be partially compensated for by increasing $\mu$, but that reduces the $\mathrm{KL}(q_{\boldsymbol{\phi}}(\mathbf{z}|\mathbf{y}), p_{\boldsymbol{\theta}}(\mathbf{z}))$ penalty, which is required to ensure $q_{\boldsymbol{\phi}_y}(\mathbf{z}|\mathbf{y})$ is a broad distribution with good coverage of the concept.

The JMVAE objective of Suzuki et al. (2017) has the form $\mathcal{L}(\boldsymbol{\theta},\boldsymbol{\phi}) = \mathbb{E}_{\hat{p}(\mathbf{x},\mathbf{y})}\left[J(\mathbf{x},\mathbf{y},\boldsymbol{\theta},\boldsymbol{\phi})\right]$, where

$$J(\mathbf{x},\mathbf{y},\boldsymbol{\theta},\boldsymbol{\phi}) = \mathrm{elbo}_{1,\lambda,1}(\mathbf{x},\mathbf{y},\boldsymbol{\theta},\boldsymbol{\phi}) - \alpha\left[\mathrm{KL}(q_{\boldsymbol{\phi}}(\mathbf{z}|\mathbf{x},\mathbf{y}), q_{\boldsymbol{\phi}_y}(\mathbf{z}|\mathbf{y})) + \mathrm{KL}(q_{\boldsymbol{\phi}}(\mathbf{z}|\mathbf{x},\mathbf{y}), q_{\boldsymbol{\phi}_x}(\mathbf{z}|\mathbf{x}))\right]$$

At first glance, forcing $q_{\boldsymbol{\phi}}(\mathbf{z}|\mathbf{y})$ to be close to $q_{\boldsymbol{\phi}}(\mathbf{z}|\mathbf{x},\mathbf{y})$ seems undesirable, since the latter will typically be close to a delta function, since there is little posterior uncertainty in $\mathbf{z}$ once we see the image $\mathbf{x}$. However, in Appendix A.1, we use results from Hoffman & Johnson (2016) to show that $\mathbb{E}_{\hat{p}(\mathbf{x},\mathbf{y})}\left[\mathrm{KL}(q_{\boldsymbol{\phi}}(\mathbf{z}|\mathbf{x},\mathbf{y}), q_{\boldsymbol{\phi}_y}(\mathbf{z}|\mathbf{y}))\right]$ can be written in terms of $\mathrm{KL}(q_{\boldsymbol{\phi}}^{\mathrm{avg}}(\mathbf{z}|\mathbf{y}), q_{\boldsymbol{\phi}_y}(\mathbf{z}|\mathbf{y}))$, where $q_{\boldsymbol{\phi}}^{\mathrm{avg}}(\mathbf{z}|\mathbf{y}) = \mathbb{E}_{\hat{p}(\mathbf{x}|\mathbf{y})}\left[q_{\boldsymbol{\phi}}(\mathbf{z}|\mathbf{x},\mathbf{y})\right]$ is the aggregated posterior over $\mathbf{z}$ induced by all images $\mathbf{x}$ which are associated with description $\mathbf{y}$. This ensures that $q_{\boldsymbol{\phi}_y}(\mathbf{z}|\mathbf{y})$ will cover the embeddings of *all* the images associated with concept $\mathbf{y}$. However, since there is no $\mathrm{KL}(q_{\boldsymbol{\phi}_y}(\mathbf{z}|\mathbf{y}), p_{\boldsymbol{\theta}}(\mathbf{z}))$ term, the diversity of the samples is slightly reduced for novel concepts compared to TELBO, as we show empirically in Section 5. On the flip side, the benefit of using the aggregated posterior to fit the $q(\mathbf{z}|\mathbf{y})$ inference network is that one can expect sharper images, as this ensures we will sample $\mathbf{z} \sim q(\mathbf{z}|\mathbf{y})$ which have been seen by the image decoder $p_{\boldsymbol{\theta}}(\mathbf{x}|\mathbf{z})$ during joint training. If the aggregated posterior does not exactly match the prior (which is known to happen in VAE-type models, see Hoffman & Johnson (2016)) then regularizing with respect to the prior (as TELBO does) can generate samples in parts of space not seen by the image decoder, which can potentially lead to less "correct" samples. Again, our empirical findings in Section 5 confirm this tradeoff between correctness and coverage implicit in choices of TELBO *vs.* JMVAE.

The SCAN method of Higgins et al. (2017b) first fits a standard $\beta$-VAE model (Higgins et al., 2017a) on unlabeled images (or rather, features derived from images using a pre-trained denoising autoencoder) by maximizing $\mathcal{L}(\boldsymbol{\theta}_x,\boldsymbol{\phi}_x) = \mathbb{E}_{\hat{p}(\mathbf{x})}\left[\mathrm{elbo}_{1,\beta_x}(\mathbf{x},\boldsymbol{\theta}_x,\boldsymbol{\phi}_x)\right]$. They then fit a second VAE by maximizing $\mathcal{L}(\boldsymbol{\theta}_y,\boldsymbol{\phi}_y) = \mathbb{E}_{\hat{p}(\mathbf{x},\mathbf{y})}\left[J(\mathbf{x},\mathbf{y},\boldsymbol{\theta}_y,\boldsymbol{\phi}_y,\boldsymbol{\phi}_x)\right]$, where

$$J(\mathbf{x},\mathbf{y},\boldsymbol{\theta}_y,\boldsymbol{\phi}_y,\boldsymbol{\phi}_x) = \mathrm{elbo}_{1,\beta_y}(\mathbf{y},\boldsymbol{\theta}_y,\boldsymbol{\phi}_y) - \alpha\mathrm{KL}(q_{\boldsymbol{\phi}_x}(\mathbf{z}|\mathbf{x}), q_{\boldsymbol{\phi}_y}(\mathbf{z}|\mathbf{y}))$$

This is very similar to JMVAE, since $q_{\boldsymbol{\phi}_x}(\mathbf{z}|\mathbf{x}) \approx q_{\boldsymbol{\phi}}(\mathbf{z}|\mathbf{x},\mathbf{y})$, when $(\mathbf{x},\mathbf{y})$ is a matching pair of images and labels. An important difference, however, is that SCAN treats the attribute vectors $\mathbf{y}$ as atomic symbols; this has the advantage that there is no need to handle missing inputs, but the disadvantage that they cannot infer the meaning of unseen attribute combinations at test time, unless they are "taught" them by having them paired with images. Also, they rely on $\beta_x > 1$ as a way to get compositionality, assuming that a disentangled latent space will suffice. However, in Appendix A.3, we show that unsupervised learning of the latent space given images alone can result in poor results when some of the attributes in the compositional concept hierarchy are non-visual, such as parity of an MNIST digit. Our approach always takes the labels into consideration when learning the latent space, permitting well-organized latent spaces even in the presence of non-visual concepts (c.f. the difference between PCA and LDA).

**Handling missing inputs.** Conditional generative models of images, of the form $p(\mathbf{x}|\mathbf{y})$, have problems with missing input attributes, as do inference networks $q(\mathbf{z}|\mathbf{y})$ for VAEs. Hoffman (2017) uses MCMC to fit a latent Gaussian model, which can in principle handle missing data; however, he initializes the Markov chain with the posterior mode computed by an inference network, which cannot easily handle missing inputs. One approach we can use, if we have a joint model, is to estimate or impute the missing values, as follows: $\hat{\mathbf{y}} = \arg\max_{\mathbf{y}_{\mathcal{M}}} p(\mathbf{y}_{\mathcal{M}}|\mathbf{y}_{\mathcal{O}})$, where $p(\mathbf{y}_{\mathcal{M}},\mathbf{y}_{\mathcal{O}})$ models dependencies between attributes. We can then sample images using $p(\mathbf{x}|\hat{\mathbf{y}})$. This approach was

used in Yan et al. (2016) to handle the case where some of the pixels being passed into an inference network were not observed. However, conditioning on an imputed value will give different results from not conditioning on the missing inputs; only the latter will increase the posterior uncertainty in order to correctly represent less precise concepts with broader support.

**Gaussian embeddings.** There are many papers that embed images and text into points in a vector space. However, we want to represent concepts of different levels of abstraction, and therefore want to map images and text to regions of latent space. There are some prior works that use Gaussian embeddings for words (Vilnis & McCallum, 2015; Athiwaratkun & Wilson, 2017), sometimes in conjunction with images (Mukherjee & Hospedales, 2016; Ren et al., 2016). Our method differs from these approaches in several ways. First, we maximize the likelihood of $(\mathbf{x}, \mathbf{y})$ pairs, whereas the above methods learn a Gaussian embedding using a contrastive loss. Second, our PoE formulation ensures that the covariance of the posterior $q(\mathbf{z}|\mathbf{y}_\mathcal{O})$ is adaptive to the data that we condition on. In particular, it becomes narrower as we observe more attributes (because the precision matrices sum up), which is a property not shared by other embedding methods.

**Abstraction and compositionality.** Young et al. (2014) represent the extension of a concept (described by a noun phrase) in terms of a set of images whose captions match the phrase. By contrast, we use a parametric probability distribution in a latent space that can generate new images. Vendrov et al. (2016) use order embeddings, where they explicitly learn subsumption-like relationships by learning a space that respects a partial order. In contrast, we reason about generality of concepts via the uncertainty induced by their latent representation. There has been some work on compositionality in the language/vision literature (see e.g., Atzmon et al. (2016); Johnson et al. (2017); Agrawal et al. (2017)), but none of these papers use generative models, which is arguably a much more stringent test of whether a model has truly "understood" the meaning of the components which are being composed.

## 5 EXPERIMENTAL RESULTS

In this section, we fit the JVAE model to two different datasets (MNIST-A and CelebA), using the TELBO objective, as well as BiVCCA and JMVAE. We measure the quality of the resulting model using the 3 C's, and show that our method of handling missing data behaves in a qualitatively reasonable way.

### 5.1 MNIST-A

**Dataset.** In this section, we report results on the MNIST-A dataset. This is created by modifying the original MNIST dataset as follows. We first create a compositional concept hierarchy using 4 discrete attributes, corresponding to class label (10 values), location (4 values), orientation (3 values), and size (2 values). Thus there are `10x2x3x4=240` unique concepts in total. We then sample $\sim 290$ example images of each concept, and create both an iid and compositional split of the data. See Appendix A.2 for details.

**Models and algorithms.** We train the JVAE model on this dataset using TELBO, BiVCCA and JMVAE objectives. We use Adam (Kingma & Ba, 2015) for optimization, with a learning rate of 0.0001, and a minibatch size of 64. We train all models for 250,000 steps (we generally found that the models do not tend to overfit in our experiments). Our models typically take around a day to train on NVIDIA Titan X GPUs. For the image models, $p(\mathbf{x}|\mathbf{z})$ and $q(\mathbf{z}|\mathbf{x})$, we use the DCGAN architecture from Radford et al. (2016). Our generated images are of size 64×64, as in Radford et al. (2016). For the attribute models, $p(y_k|\mathbf{z})$ and $q(\mathbf{z}|y_k)$, we use MLPs. For the joint inference network, $q(\mathbf{z}|\mathbf{x}, \mathbf{y})$, we use a CNN combined with an MLP. We use $d = 10$ latent dimensions for all models. We choose the hyperparameters for each method so as to maximize JS-overall, which is an overall measure of correctness and coverage (see Section 3) on a validation set of attribute queries. See Appendix A.4 for further details on the model architectures.

**Evaluation.** To measure correctness and coverage, we first train the observation classifier on the full iid dataset, where it gets to an accuracy of 91.18% for class label, 90.56% for scale, 92.23% for

**Query:** 0, small, clockwise, top-right

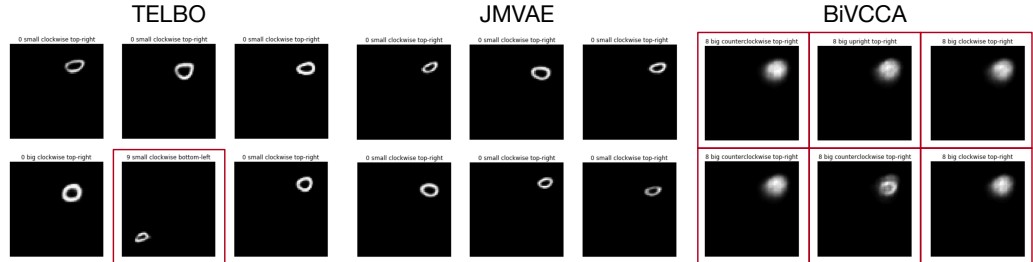

Figure 3: Samples from attribute vectors seen at training time, generated by the 3 different models. We plot the posterior mean of each pixel, $\mathbb{E}[\mathbf{x}|\mathbf{z}_s]$, where $\mathbf{z}_s \sim q_{\phi_y}(\mathbf{z}|\mathbf{y})$. The caption at the top of each little image is the predicted attribute values. The border of the generated image is red if any of the attributes are predicted incorrectly. (The observation classifier is fed sampled images, not the mean image that we are showing here.)

orientation, and 100% for location. Consequently, it is a reliable way to assess the quality of samples from various generative models (see Appendix A.5 for details). We then compute correctness and coverage on the iid dataset, and coverage on the comp dataset.

**Familiar concrete concepts.** We start by assessing the quality of the models in the simplest setting, which is where the test concepts are fully specified (i.e., all attributes are known), and the concepts have been seen before in the training set (i.e., we are using the iid split). Figure 4a shows the correctness scores for the three methods. (Since the test concepts are fully grounded, coverage is not well defined, since there are no missing attributes.) We see that TELBO has a correctness of 82.08%, which is close to that of JMVAE (85.15%); both methods significantly outperform BiVCCA (67.38%). To gain more insight, Figure 3 shows some samples from each of these methods for a leaf concept chosen at random. We see that the images generated by BiVCCA are very blurry, for reasons we discussed in Section 4. Note that these blurry images are correctly detected by the attribute classifier.[3] We also see that the JMVAE samples all look good (in this example). Most of the samples from TELBO are also good, although there is one error (correctly detected by the attribute classifier).

| Method | #Attributes | Coverage (%) | Correctness (%) |
|---|---|---|---|
| | | iid split | |
| TELBO | | - | $82.08 \pm 0.56$ |
| JMVAE | 4 | - | $85.15 \pm 0.26$ |
| BiVCCA | | - | $67.38 \pm 0.69$ |
| TELBO | | $91.14 \pm 0.53$ | $81.63 \pm 0.38$ |
| JMVAE | 3 | $88.52 \pm 0.37$ | $82.00 \pm 0.37$ |
| BiVCCA | | $85.28 \pm 0.68$ | $70.68 \pm 0.87$ |
| TELBO | | $90.32 \pm 0.57$ | $82.03 \pm 1.37$ |
| JMVAE | 2 | $87.89 \pm 0.69$ | $81.02 \pm 1.05$ |
| BiVCCA | | $85.09 \pm 0.76$ | $72.33 \pm 2.31$ |
| TELBO | | $90.94 \pm 0.19$ | $83.67 \pm 1.70$ |
| JMVAE | 1 | $88.70 \pm 0.35$ | $81.58 \pm 1.78$ |
| BiVCCA | | $85.53 \pm 0.27$ | $68.36 \pm 2.21$ |
| | | Compositional split | |
| TELBO | | - | $75.61 \pm 1.43$ |
| JMVAE | 4 | - | $76.86 \pm 1.30$ |
| BiVCCA | | - | $68.58 \pm 1.02$ |

(a) Evaluation of different approaches on the test set. Higher numbers are better. We report standard deviation across 5 splits of the test set.

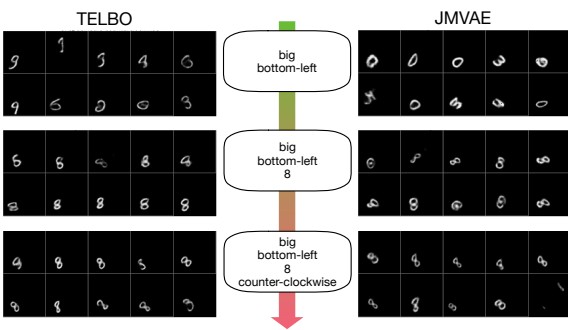

(b) Mean images generated by TELBO and JMVAE in response to queries at different levels of abstraction, starting from abstract (top) to refined (bottom).

Figure 4: **(a)** We show quantitaive results on the 3C's on MNIST-A. **(b)** Qualitative results on MNIST-A for various queries. For refined/fully specified queries, we can see that both TELBO and JMVAE produce good correctness, *i.e.*, the images produced follow constraints placed by the specified attributes. When the attribute 'orientation' is unspecified, we see that TELBO produces upright and counter clockwise digits, while JMVAE produces clockwise and upright digits. Finally, when we leave the digit unspecified (top), we see that TELBO appears to generate a more diverse set of digits (9, 3, 8, 6) while JMVAE produces 0 and 3.

[3] We chose the value of $\mu = 0.7$ based on maximizing correctness score on the validation set. Nevertheless, this does not completely eliminate blurriness, as we can see.

**Novel abstract concepts.** Next we assess the quality of the models when the test concepts are abstract, i.e., one or more attributes are not specified. (Note that the model was never trained on such abstract concepts.) Figure 4a shows that the correctness scores for JMVAE seems to drop somewhat (from about 85% to about 81.5%), although it remains steady for TELBO and BiVCCA. We also see that the coverage of TELBO is higher than the other methods, due to the use of the $\mathrm{KL}(q_{\phi_y}(\mathbf{z}|\mathbf{y}), p_{\boldsymbol{\theta}}(\mathbf{z}))$ regularizer, as we discussed in Section 4. Figure 4b illustrates how the methods respond to concepts of different levels of abstraction. The samples from the TELBO seem to be more diverse, which is consistent with the numbers in Figure 4a.

**Compositionally novel concrete concepts.** Finally we assess the quality of the models when the test concepts are fully specified, but have not been seen before (i.e., we are using the comp split). Figure 4a shows some quantitative results. We see that the correctness for TELBO and JMVAE has dropped from about 82% to about 75%, since this task is much harder, and requires "strong generalization". However, as before, we see that both TELBO and JMVAE outperform BiVCCA, which has a correctness of about 69%. See Appendix A.7 qualitative results and more details.

## 5.2 CELEBA

In this section, we report results on the CelebA dataset (Liu et al., 2015). In particular, we use the version that was used in Perarnau et al. (2016), which selects 18 visually distinctive attributes, and generate images of size 64×64; see Appendix A.8 for more details on the CelebA dataset and Appendix A.4 for details of the model architectures. Figure 5 shows some sample qualitative results. On the top left, we show some images which were generated by the three methods given the concept shown in the left column. TELBO and JMVAE generate realistic and diverse images. That is, the generated images are generally of males, with mouth slightly open and smiling attributes present in the images. On the other hand, BiVCCA just generates the mean image. On the bottom left, we show what happens when we drop some attributes, thus specifying more abstract concepts. We see that when we drop the gender, we get a mixture of both male and female images for both TELBO and JMVAE. Going further, when we drop the "smiling" attribute, we see that the samples now comprise of people who are smiling as well as not smiling, and we see a mixture of genders in the samples. Further, while we see a greater diversity in the samples, we also notice a slight drop in image quality (presumably because none of the approaches has seen supervision with just 'abstract' concepts). See Appendix A.9 for more qualitative examples on CelebA. On the top right, we show some examples of visual imagination, where we ask the models to generate images from the concept "bald female", which does not occur in the training set.[4] (We omit the results from BiVCCA, which are uniformly poor.) We see that both TELBO and JMVAE can sometimes do a fairly reasonable job (although these are admittedly cherry picked results). Finally, the bottom right illustrates an interesting bias in the dataset: if we ask the model to generate images where we do not specify the value of the eyeglasses attribute, nearly all of the samples fail to included glasses, since the prior probability of this attribute is rare (about 6%).

## 6 CONCEPT NAMING WITH IMAGINATION MODELS

In this section, we demonstrate initial results which show that our imagination models can be used for concept naming, where the task is to assign a label to a set of images illustrating the concept depicted by the images. A similar problem has been studied in previous work such as Tenenbaum (1999) and Jia et al. (2013). Tenenbaum (1999) studies a set naming problem with integers (instead of images), and show that construct a likelihood function given a hypothesis set that can capture notions of the minimal/smallest hypothesis that explains the observed samples in the set. Jia et al. (2013) extend this approach to concept-naming on images, incorporating perceptual uncertainty (in recognizing the contents of an image) using a confusion matrix weighted likelihood term. While this approach first extracts labels for each image and then performs concept naming, here we test how well our generative model itself is able to generalize to concept naming without ever performing explicit classification on the images.

---

[4] There are 9 examples in the training set with the attributes (male=0, bald=1), but these turn out to all be labeling errors, as we shown in Appendix A.8.

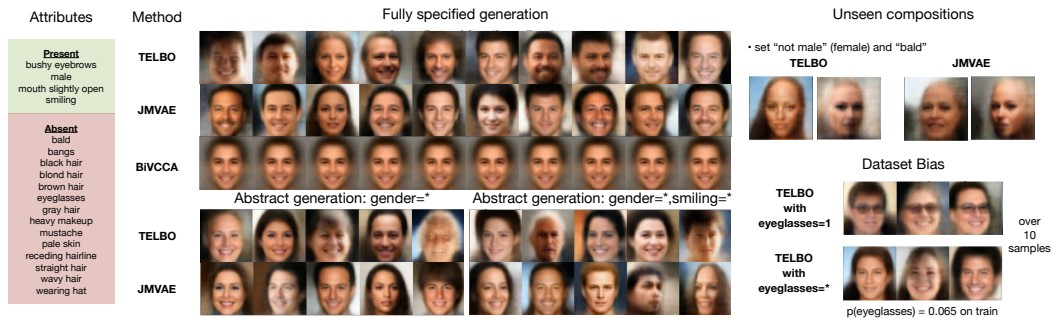

Figure 5: Sample CelebA results. Left: we show the attributes specified to be present or absent when generating images. Middle: we show 10 samples each generated from TELBO, JMVAE and BiVCCA. We see that TELBO and JMVAE genreate better samples than BiVCCA which collapses to the mean. Middle, bottom: We show five samples from TELBO and JMVAE in response to queries with unspecified attributes, and see that both approaches generate a mix in the samples, generalizing meaningfully across unspecified attributes.

In more detail, the problem setup in concept naming is as follows: we are given as input a set $\mathcal{X}$ of images, each of which corresponds to a concept in the compositional abstraction hierarchy Figure 1. The task is to assign a label $\mathbf{y} \in \mathcal{Y}$ to the set of images. One of the key challenges in concept learning is to understand "how far" to generalize in the concept hierarchy given a limited number of positive examples (Tenenbaum, 1999). That is, given a small set of images with 7 in the top-left corner and bottom-right corner, one must infer that the concept is "7" as opposed to "7, top-left". In other words, we wish to find the least common ancestor (in the concept hierarchy) corresponding to all the images in the set, given any number of images in the set, so that we can be consistent with the set. We consider two heuristic solutions to this problem:

1. **Concept-NB**: In this approach we compute $\arg\max_{\mathbf{y}} p(\mathbf{y}|\mathcal{X})$, where $p(\mathbf{y}|\mathcal{X})$ is computed using the naive bayes assumption:

$$p(\mathbf{y}|\mathcal{X}) \propto p(\mathbf{y})\Pi_{\mathbf{x}_n \in \mathcal{X}} p(\mathbf{x}_n|\mathbf{y}) = p(\mathbf{y})\Pi_{\mathbf{x}_n \in \mathcal{X}} \int d\mathbf{z}_n p(\mathbf{x}_n|\mathbf{z}_n)q(\mathbf{z}_n|\mathbf{y})$$

where $p(y)$ is chosen to be uniform across all concepts, and the integrals are approximated using Monte Carlo.

2. **Concept-Latent**: In this approach, instead of working in the observed space, we work in the latent space. That is, we pick $\arg\min_{\mathbf{y}} \mathrm{KL}(q(\mathbf{z}|\mathcal{X})|q(\mathbf{z}|\mathbf{y}))$, where $q(\mathbf{z}|\mathcal{X})$ is approximated using $\sum_{\mathbf{x} \in \mathcal{X}} q(\mathbf{z}|\mathbf{x})$, which is a mixture of gaussians. The KL divergence can be computed analytically by considering the first two moments of the gaussian mixture[5].

## 6.1 Experimental Setup

We use the MNIST-A dataset for the concept naming studies. We consider the fully specified attribute labels in the MNIST-A hierarchy, and consider differrent patterns of missingness (corresponding to different nodes in the abstraction hierarchy) by dropping attributes. Specifically, we ignore the case where no attribute is specified, and consider a uniform distribution over the rest of the $(2^4 - 1 = 15)$ patterns of missingness. Now, for each fully specified attribute pattern in the iid split of MNIST-A, we sample four missingness patterns and repeat across all fully specified attributes to form a bank of 960 candidate names that a model must choose. We randomly select three subsets of 100 candidate names (and the corresponding images) to form the query set for concept naming, namely tuples of $(\mathbf{y}, \mathcal{X})$. Specifically, given all the images in the eval set for a concept $\mathbf{y}$, we form $\mathcal{X}$ using a randomly sampled subset of 5 images. We report the accuracy metric, measuring how often the selected concept for a set $\mathcal{X}$ matches the ground truth concept, across three different splits of 100 datapoints.

---

[5]Given a Gaussian mixture of the form $g(\mathbf{x}) = \sum_i \pi_i f(\mathbf{x}; \mu_i, \sigma_i)$, where $f$ is the pdf for the Gaussian distribution, the first order moment, that is, the mean of $g(\mathbf{x})$ is given by: $\sum_i \pi_i \mu_i$. The variance is given by: $\sum_i \pi_i \sigma_i^2 + \sum_i \pi_i \mu_i^2 - (\sum_i \pi_i \mu_i)^2$.

| Approach | Concept-Latent (%) | Concept-NB (%) |
|---|---|---|
| TELBO | $35.66 \pm 2.05$ | $17.66 \pm 1.70$ |
| JMVAE | $54.66 \pm 4.92$ | $13.33 \pm 2.05$ |
| BiVCCA | $28.00 \pm 4.54$ | $18.00 \pm 1.40$ |
| Random | $0.28 \pm 0.00$ | $0.28 \pm 0.00$ |
| Most Frequent | $6.33 \pm 1.88$ | $6.33 \pm 1.88$ |

Table 1: Accuracy of Imagination models on Concept Naming. Higher is better.

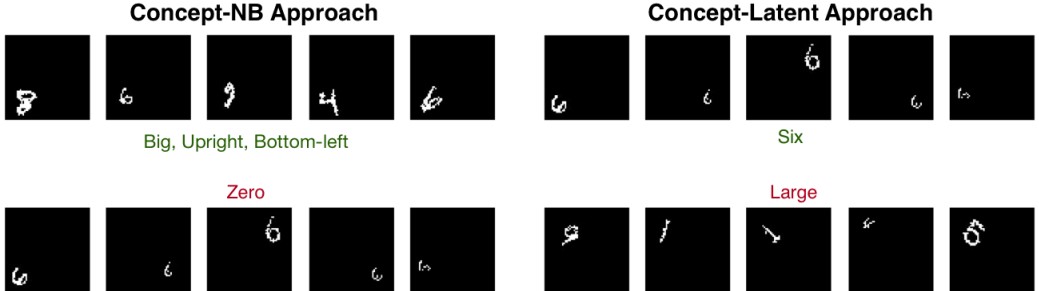

Figure 6: A qualitative illustration of some of the examples from concept naming models. Top-left: an example of a sample that is correctly named by a Concept-NB model. However, the Concept-NB model is not that strong and often gets simple concepts such as digits incorrect, making mistakes between 6 and 0, for example (bottom-left). This is likely because the only way in which the Concept-NB approach reasons about the set is not via a "meaningful" low dimensional latent variable but via a sampling distribution on a high dimensional space of images. The Concept-Latent model is able to do better on the same set of images, and classify the set as the concept "6". Finally, we show a failure case of the model where it incorrectly classifies the digits as being large (there is a small digit in the set), and ignores the fact that all of the digits are in the top-left.

## 6.2 RESULTS

We evaluate the best versions of TELBO, JMVAE, and BiVCCA on the iid split of MNIST-A for concept naming (Table 1). In general, we find that Concept-NB approaches perform significantly worse than Concept-Latent approaches. For example, the best Concept-NB approach (using TELBO/BiVCCA objective) gets to an accuracy of around $18\%$, while Concept-Latent using JMVAE gets to $54.66 \pm 4.92\%$. In general, these numbers are better than a random chance baseline which would get to $0.28\%$ (picking one of 348 effective options, after collating the 960 candidate names based on missingness patterns), while picking the most frequent (ground truth) fully-specified $\mathbf{y}$ depicted across an image set gets to $6.33 \pm 1.88\%$. Figure 6 shows some qualitative examples from Concept-NB as well as Concept-Latent models for concept / set classification. We observe that the Concept-Latent models are much more powerful than using Concept-NB in terms of naming the concept based on few positive examples from the support set.

## 7 CONCLUSIONS AND FUTURE WORK

We have shown how to create generative models which can "imagine" compositionally novel concrete and abstract visual concepts. In the future we would like to explore richer forms of description, beyond attribute vectors, such as natural language text, as well as compositional descriptions of scenes, which will require dealing with a variable number of objects.

## ACKNOWLEDGMENTS

We would like to thank Hernan Moraldo for his help in writing the JVAE library, Alex Alemi for valuable insights on TELBO and JMVAE, and Sergio Guadarrama and Harsh Satija for numerous discussions around the project. Finally we would like to thank Devi Parikh for advice on the CelebA experiments, and Stefan Lee and Yash Goyal for feedback on an initial version of this draft.

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

# A  APPENDIX

## A.1  ANALYSIS OF JMVAE OBJECTIVE

The JMVAE objective of (Suzuki et al., 2017) has the form

$$J(\mathbf{x}, \mathbf{y}, \boldsymbol{\theta}, \boldsymbol{\phi}) = \text{elbo}(\mathbf{x}, \mathbf{y}, \boldsymbol{\theta}, \boldsymbol{\phi}) - \alpha \left[ \text{KL}(q_{\boldsymbol{\phi}}(\mathbf{z}|\mathbf{x}, \mathbf{y}), q_{\boldsymbol{\phi}_y}(\mathbf{z}|\mathbf{y})) + \text{KL}(q_{\boldsymbol{\phi}}(\mathbf{z}|\mathbf{x}, \mathbf{y}), q_{\boldsymbol{\phi}_x}(\mathbf{z}|\mathbf{x})) \right]$$

Let us focus on the $\text{KL}(q_{\boldsymbol{\phi}}(\mathbf{z}|\mathbf{x}, \mathbf{y})|q_{\boldsymbol{\phi}_y}(\mathbf{z}|\mathbf{y}))$ term. Let $\mathcal{Y}$ be the set of unique labels (attribute vectors) in the training set, $\mathcal{X}_i$ be the indices of the images associated with label $\mathbf{y}_i$, and let $N_i = |\mathcal{X}_i|$ be the size of that set. Then we can write

$$\mathbb{E}_{\hat{p}(\mathbf{x}, \mathbf{y})} \left[ \text{KL}(q_{\boldsymbol{\phi}}(\mathbf{z}|\mathbf{x}, \mathbf{y})|q_{\boldsymbol{\phi}_y}(\mathbf{z}|\mathbf{y})) \right] = \frac{1}{|\mathcal{Y}|} \sum_{i \in \mathcal{Y}} \frac{1}{N_i} \sum_{n \in \mathbf{X}_i} \text{KL}(q_{\boldsymbol{\phi}}(\mathbf{z}|\mathbf{x}_n, \mathbf{y}_i), q_{\boldsymbol{\phi}_y}(\mathbf{z}|\mathbf{y}_i)) \quad (3)$$

As explained in (Hoffman & Johnson, 2016), we can rewrite this by treating the index $n \in \{1, \cdots, N_i\}$ as a random variable, with prior $q(n|\mathbf{y}_i) = 1/N_i$. Also, let us define the likelihood $q(\mathbf{z}|n, \mathbf{y}_i) = q_{\boldsymbol{\phi}}(\mathbf{z}|\mathbf{x}_n, \mathbf{y}_i)$. Using this notation, we can show that the above average KL becomes

$$\frac{1}{|\mathcal{Y}|} \sum_{i \in \mathcal{Y}} \left\{ \text{KL}(q_{\boldsymbol{\phi}}^{\text{avg}}(\mathbf{z}|\mathbf{y}_i)|q_{\boldsymbol{\phi}_y}(\mathbf{z}|\mathbf{y}_i)) + \log N_i - \mathbb{E}_{q_{\boldsymbol{\phi}_y}(\mathbf{z}|\mathbf{y}_i)} \left[ \mathbb{H}(q(n|\mathbf{z}, \mathbf{y}_i)) \right] \right\} \quad (4)$$

where

$$q_{\boldsymbol{\phi}}^{\text{avg}}(\mathbf{z}|\mathbf{y}_i) = \frac{1}{N_i} \sum_{n \in \mathcal{X}_i} q_{\boldsymbol{\phi}}(\mathbf{z}|\mathbf{x}_n, \mathbf{y}_i)$$

is the average of the posteriors for that concept, and $q(n|\mathbf{z}, \mathbf{y}_i)$ is the posterior over the indices for all the possible examples from the set $\mathcal{X}_i$, given that the latent code is $\mathbf{z}$ and the description is $\mathbf{y}_i$.

The $\text{KL}(q_{\boldsymbol{\phi}}^{\text{avg}}(\mathbf{z}|\mathbf{y}_i)|q_{\boldsymbol{\phi}_y}(\mathbf{z}|\mathbf{y}_i))$ term in Equation (4) tells us that JMVAE encourages the inference network for descriptions, $q_{\boldsymbol{\phi}_y}(\mathbf{z}|\mathbf{y}_i)$, to be close to the average of the posteriors induced by each of the images $\mathbf{x}_n$ associated with $\mathbf{y}_i$. Since each $q_{\boldsymbol{\phi}}(\mathbf{z}|\mathbf{x}_n, \mathbf{y}_i)$ is close to a delta function (since there is little posterior uncertainty when conditioning on an image), we are essentially requiring that $q_{\boldsymbol{\phi}}(\mathbf{z}|\mathbf{y}_i)$ cover the embeddings of each of these images.

## A.2  DETAILS ON THE MNIST-A DATASET

We created the MNIST-A dataset as follows. Given an image in the original MNIST dataset, we first sample a discrete scale label (big or small), an orientation label (clockwise, upright, and anti-clockwise), and a location label (top-left, top-right, bottom-left, bottom-right).

Next, we converted this vector of discrete attributes into a vector of continuous transformation parameters, using the procedure described below:

- **Scale:** For big, we sample scale values from a Gaussian centered at 0.9 with a standard deviation of 0.1, while for small we sample from a Gaussian centered at 0.6 with a standard deviation of 0.1. In all cases, we reject and draw a sample again if we get values outside the range $[0.4, 1.0]$, to avoid artifacts from upsampling or problems with illegible (small) digits.

- **Orientation:** For the clockwise label, we sample the amount of rotation to apply for a digit from a Gaussian centered at +45 degrees, with a standard deviation of 10 degrees. For anti-clockwise, we use a Gaussian at -45 degrees, with a standard deviation of 10 degrees. For upright, we set the rotation to be 0 degrees always.

- **Location:** For location, we place Gaussians at the centers of the four quadrants in the image, and then apply an offset of `image_size/16` to shift the centers a bit towards the corresponding corners. We then use a standard deviation of `image_size/16` and sample locations for centers of the digits. We reject and draw the sample again if we find that the location for the center would place the extremities of the digit outside of the canvas.

Finally, we generate the image as follows. We first take an empty black canvas of size `64x64`, rotate the original `28x28` MNIST image, and then scale and translate the image and paste it on

the canvas. (We use bicubic interpolation for scaling and resizing the images.) Finally, we use the method of (Salakhutdinov & Murray, 2008) to binarize the images. See Figure 7 for example images generated in this way.

We repeat the above process of sampling labels, and applying corresponding transformations, to generate images 10 times for each image in the original MNIST dataset. Each trial samples labels from a uniform categorical distribution over the sample space for the corresponding attribute. Thus, we get a new MNIST-A dataset with 700,000 images from the original MNIST dataset of 70,000 images. We split the images into a train, val and test set of 85%, 5%, and 10% of the data respectively to create the IID split. To create the compositional split, we split the `10x2x3x4=240` possible label combinations by the sample train/val/test split, giving us splits of the dataset with non-overlapping label combinations.

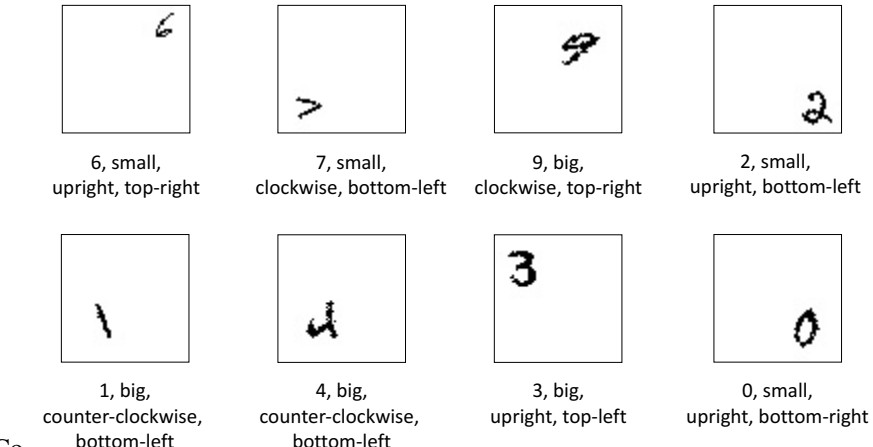

Figure 7: Example binary images from our MNIST-A dataset.

### A.3 $\beta$-VAE vs. JOINT VAE

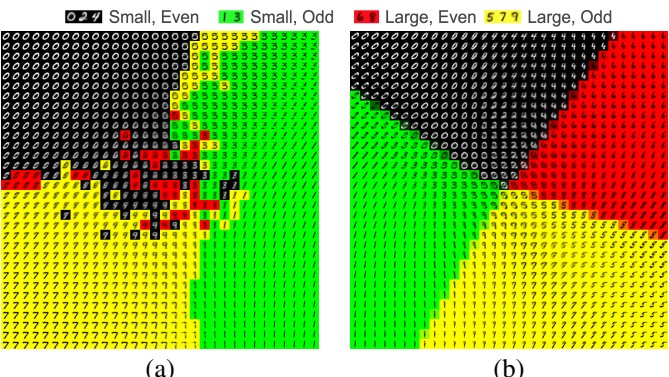

Figure 8: Visualization of the benefit of semantic annotations for learning a good latent space. Each small digit is a single sample generated from $p(x|z)$ from the corresponding point $z$ in latent space. (a) $\beta$-VAE fit to images without annotations. The color of a point $z$ is inferred from looking at the attributes of the training image that maps to this point of space using $q(z|x)$. Note that the red region (corresponding to the concept of large and even digits) is almost non existent. (b) Joint-VAE fit to images with annotations. The color of a point $z$ is inferred from $p(y|z)$.

$\beta$-VAE Higgins et al. (2017a) is an approach that aims to learn disentangled latent spaces. It does this by modifying the ELBO objective, so that it scales the $\mathrm{KL}(q(\mathbf{z}|\mathbf{x}), p(\mathbf{z}))$ term by a factor $\beta > 1$. This gives rise to disentangled spaces since the prior $p(\mathbf{z}) = \mathcal{N}(\mathbf{z}|\mathbf{0}, \mathbf{I})$ is factorized (see (Achille & Soatto, 2017) for details). However, to learn latent spaces that correspond to high level concepts, this is not sufficient: we need to use labeled data as well.

To illustrate this, we set up an experiment where we learn a 2d latent space for standard MNIST digit images, but where we replace the label with two binary attributes: parity (odd *vs.*even) and magnitude (value $< 5$ or $>= 5$). We call this dataset MNIST-2bit.

In Figure 8(a), we show the results of fitting a 2d $\beta$-VAE model (Higgins et al., 2017a) to the images in MNIST-2bit, *ignoring the attributes*. We perform a hyperparameter sweep over $\beta$, and pick the one that gives the best looking latent space (this corresponds to a value of $\beta = 10$). At each point $z$ in the latent 2d space, we show a single image sampled from $p(x|z)$. To derive the colors for each point in latent space, we proceed as follows: we embed each training image $x$ (with label $y(x)$) into latent space, by computing $\hat{z}(x) = E_{q(z|x)}[z]$. We then associate label $y(x)$ with this point in space. To derive the label for an arbitrary point $z$, we lookup the closest embedded training image (using $\ell_2$ distance in $z$ space), and use its corresponding label. We see that the latent space is useful for autoencoding (since the generated images look good), but it does not capture the relevant semantic properties of parity and magnitude. In fact, we argue that there is no way of forcing the model to learn a latent space that captures such high level conceptual properties from images alone.

In Figure 8(b), we show the results of fitting a joint VAE model to MNIST-2bit, by optimizing $\text{elbo}(x, y)$ on images and attributes (*i.e.*, we do not include the uni-modality $\text{elbo}(x)$ and $\text{elbo}(y)$ terms in this experiment.) Now the color codes are derived from $p(y|z)$ rather than using nearest neighbor retrieval. We see that the latent space autoencodes well, and also captures the 4 relevant types of concepts. In particular, the regions are all convex and linearly seperable, which facilitates the learning of a good imagination function $q(z|y)$, interpolation, retrieval, and other latent-space tasks.

A skeptic might complain that we have created an arbitrary partitioning of the data, that is unrelated to the appearance of the objects, and that learning such concepts is therefore "unnatural". But consider an agent interacting with an environment by touching digits on a screen. Suppose the amount of reward they get depends on whether the digit that they touch is small or big, or odd or even. In such an environment, it would be very useful for the agent to structure its internal representation to capture the concepts of magnitude and parity, rather than in terms of low level visual similarity. (In fact, (Scarf et al., 2011) showed that pigeons can learn simple numerical concepts, such as magnitude, by rewarding them for doing exactly this!) Language can be considered as the realization of such concepts, which enables agents to share useful information about their common environments more easily.

## A.4 Details of the neural network architectures

As explained in the main paper, we fit the joint graphical model $p(x, y, z) = p(z)p(x|z)p(y|z)$ with inference networks $q(z|x, y)$, $q(z|x)$, and $q(z|y)$. Thus, our overall model is made up of three encoders (denoted with $q$) and two decoders (denoted with $p$). Across all models we use the exponential linear unit (ELU) which is a leaky non-linearity often used to train VAEs. We explain the architectures in more detail below.

**MNIST-A model architecture**

- Image decoder, $p(x|z)$: Our architecture for the image decoder exactly follows the standard DCGAN architecture from (Radford et al., 2016), where the input to the model is the latent state of the VAE.

- Label decoder, $p(y|z)$: Our label decoder assumes a factorized output space $p(y|z) = \prod_{k \in \mathcal{A}} p(y_k|z)$, where $y_k$ is each individual attribute. We parameterize each $p(y_k|z)$ with a two-layer MLP with 128 hidden units each. We apply a small amount of $\ell_2$ regularization to the weight matrices.

- Image and Label encoder, $q(z|x, y)$: Our architecture (Figure 9) for the image-label encoder first separately processes the images and the labels, and then concatenates them downstream in the network and then passes the concatenated features through a multi-layered perceptron. More specifically, we have convolutional layers which process image into $32, 64, 128, 16$ feature maps with strides $1, 2, 2, 2$ in the corresponding layers. We use batch normalization in the convolutional layers before applying the ELU non-linearity. On the label encoder side, we first encode the each attribute label into a 32d continuous vector and then pass each individual attribute vector through a 2-layered MLP with 512 hidden dimensions each. For example, for MNIST-A we have 4 attributes, which gives us 4 vectors of 512d. We

then concatenate these vectors and pass it through a two layer MLP. Finally we concatenate this label feature with the image feature after the convolutional layers (after flattening the conv-features) and then pass the result through a 2 layer MLP to predict the mean ($\mu$) and standard deviation ($\sigma$) for the latent space gaussian. Following standard practice, we predict $\log \sigma$ for the standard deviation in order to get values which are positive.

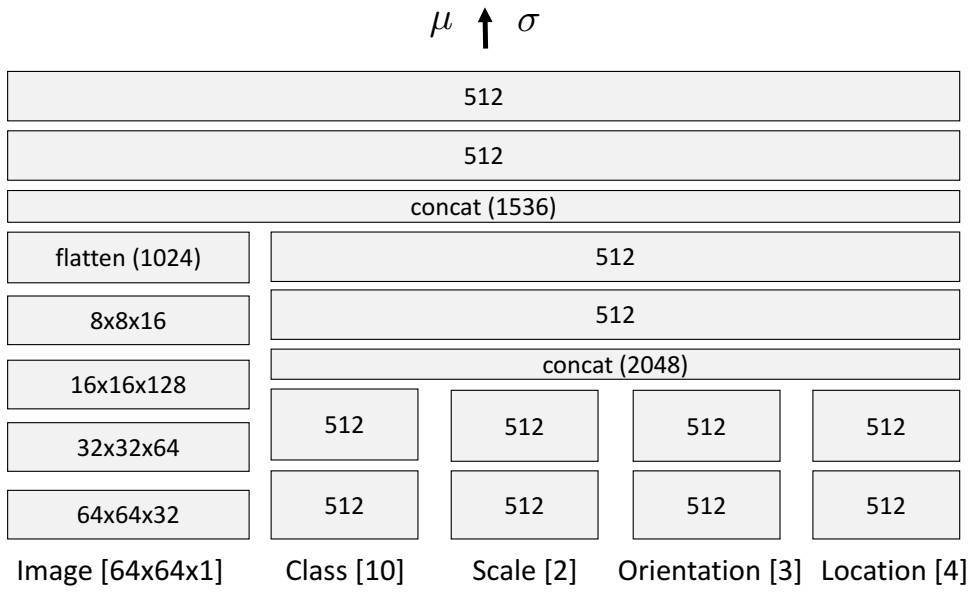

Figure 9: Architecture for the $q(z|x, y)$ network in our JVAE models for MNIST-A. Images are (64x64x1), class has 10 possible values, scale has 2 possible values, orientation has 3 possible values, and location has 4 possible values.

- Image encoder, $q(z|x)$: The image encoder (Figure 10a) uses the same architecture to process the image as the image feature extractor in $q(z|x, y)$ network described above. After the conv-features, we pass the result through a 3-layer MLP to get the latent state mean and standard deviation vectors following the procedure described above.

- Label encoder, $q(z|y)$: The label encoder (Figure 10b) part of the architecture uses the same design choices to process the labels as the label encoder part in the $q(z|x, y)$ network. After obtaining the concatenated label feature vectors, we pass the result through a 4-layered MLP with 512 hidden dimensions each and then finally obtain the mean ($\mu$) and $\log \sigma$ values for each dimension in the latent state of the VAE.

**MNIST-A Observation Classifier Model** We next describe the architecuture of the observation classifier we use for evaluating the 3C's on the MNIST-A dataset. The observation classifier is a convolutional neural network, with the first convolutional layer with filters of size 5×5, and 32 channels, followed by a 2×2 pooling layer applied with a stride of 2. This is followed by another convolutional layer with 5×5 filter size and 64 output channels. This is followed by another 2×2 pooling layer of stride 2. After this, the network has four heads (corresponding to each attribute), each of which is an MLP with a single hidden layer (of size 1024), with dropout applied to the activations. The final layer of the MLP outputs the logits for classifying each attribute into the corresponding categorical labels associated with it. We train this model from scratch on the MNIST-A dataset using stochastic gradient descent, batch size of 64 and a learning rate of $10^{-4}$.

**CelebA model architecture** Our design choices for CelebA closely mirror the models we built for MNIST-A. One primary difference is that we use a latent dimensionality of 18 in our CelebA experiments which matches the number of attributes we model. Meanwhile, the architectures of the image encoder, image decoder (*i.e.*DCGAN), are exactly identical to what is described above for MNIST-A exececpt that encoders take as input a 3-channel RGB image, while decoders produce a

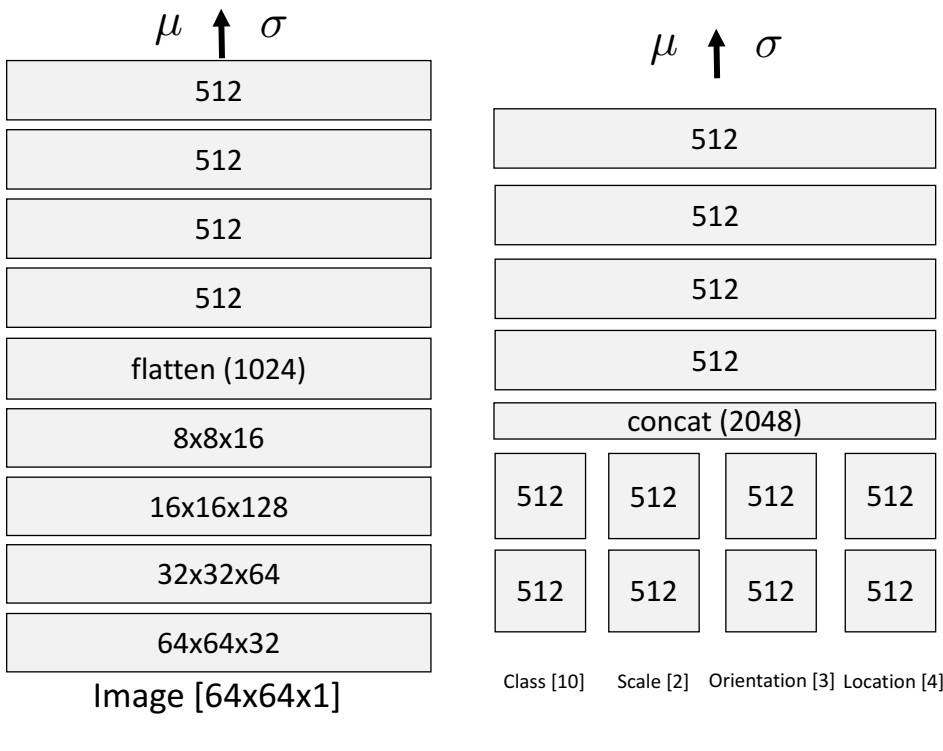

(a) Architecture for the $q(z|x)$ network.  (b) Architecture for the $q(z|y)$ network.

Figure 10: Archtectures for the single input inference networks for MNIST-A.

3-channel output. We replace the Bernoulli likelihood with Quantized Normal likelihood (which is basically gaussian likelihood with uniform noise).

In terms of the label encoder $q(z|y)$, we follow Figure 10b quite closely, except that we get as input 18 categorical (embedded) class labels as input, and we process the labels through a single hidden layer before concatenation and two hidden layers post concatenation (as opposed to two and four used in Figure 10b).

Finally, the joint encoder $q(z|x, y)$, is again based heavily on Figure 9 where we feed as input 18 labels as opposed to 4, process them through a single layer mlp of 512d, concatenate them, and then pass the result through a two hidden layer mlp of 512 d. At this point we concatenate the result with the image feature through the image feature head in Figure 9. Finally, we process the feature through another 512d single hidden layer mlp to produce the $\mu, \sigma$ values.

## A.5  OUTPUTS OF OBSERVATION CLASSIFIER ON GENERATED IMAGES

Figure 11 shows some images sampled from our TELBO model trained on MNIST-A. It also shows the attributes that are predicted by the attribute classifier. We see that the classifier often produces reasonable results that we as humans would also agree with. Thus, it acts as a reasonable proxy for humans classifying the labels for the generated images.

## A.6  HYPERPARAMTER CHOICES FOR TELBO, JMVAE, BIVCCA ON MNIST-A

We discuss more hyperparameter choices for the different objectives and how they impact performance on the MNIST-A dataset. Across all the objectives we set $\lambda_x$=1, and vary $\lambda_y$. In addition, we also discuss how the private hyperparamter choices for each loss, $\gamma$ for TELBO, $\alpha$ for JMVAE, as in Wang et al. (2016a)) and $\mu$ for BiVCCA affect performance. We use the JS-overall metric for picking hyperparameters, as explained in the main paper.

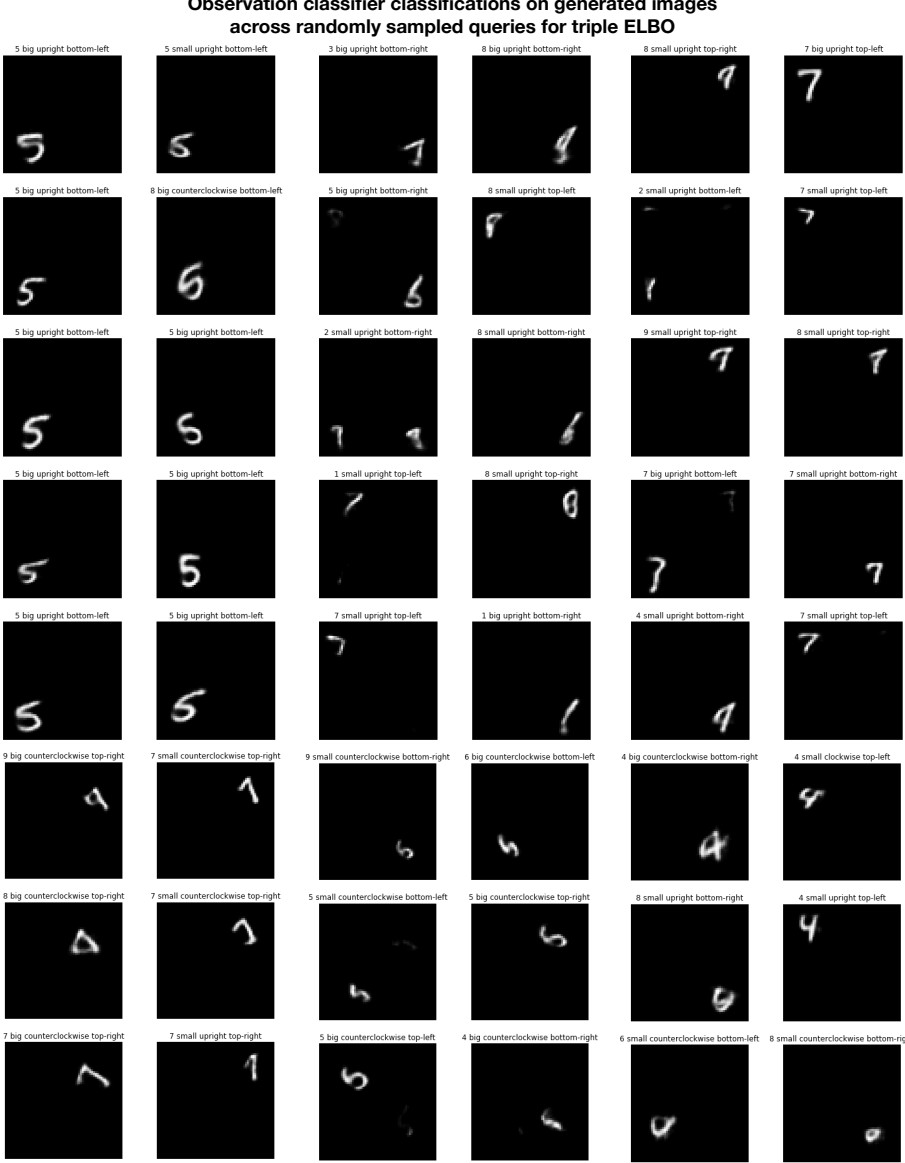

Figure 11: Randomly sampled images from the TELBO model when fed randomly sampled concepts from the iid training set. We also show the outputs of the observation classifier for the images. Note that we visualize mean images above (since they tend to be more human interpretable) but the classifier is fed samples from the model. Figure best viewed by zooming in.

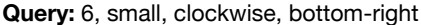

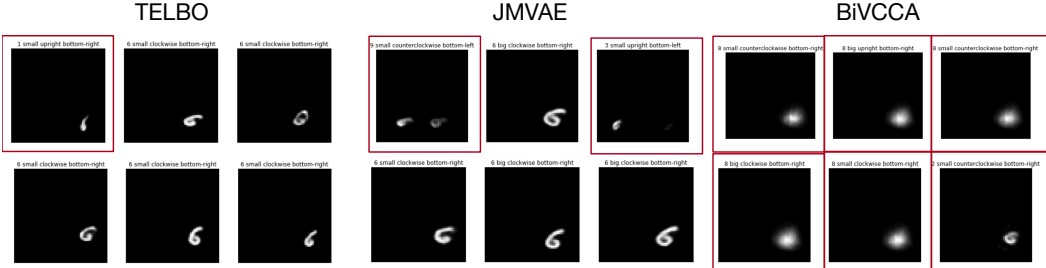

Figure 12: Compositional generalization on MNIST-A. Models are given the unseen compositional query shown at the top and each of the three columns shows the mean of the image distribution generated by the models. Images marked with a red box are those that the observation classifier detected as being incorrect. We also show the classification result from the observation classifier on top of each image. We see that TELBO and JMVAE both do really well, while BiVCCA is substantially poorer.

1. Effect of $\lambda_y$: We search for $\lambda_y$ values in the set $\{1, 50, 100\}$ for all objectives. In general, we find the setting of $\lambda_y$ in the $\mathrm{elbo}$ terms to be critical for good performance (especially on correctness). For example, at $\lambda_y$=1, we find that correctness numbers for the best performing TELBO model drop to 60.47 ($\pm$ 0.34) (from 82.08 ($\pm$ 0.56) at $\lambda_y$=50) on the validation set for $\mathtt{iid}$ queries. Similar trends can be observed for the JMVAE and BiVCCA objectives as well (with $\lambda_y$=10 being the best setting for BiVCCA, $\lambda_y$=50 for JMVAE). We have seen qualitative evidence which shows that the likelihood scaling for $\lambda_y$ affects how disentangled the latent space is along the specified attributes. When the latent space is not grouped or organized as per high-level attributes (see Figure 8 for example), the posterior distribution for a given concept is multimodal, which is hard for a gaussian inference network $q(\mathbf{z}|\mathbf{y})$ to capture. This leads to poor correctness values.

2. Effect of $\gamma$: In addition to the $\lambda_y$ scaling term which is common across all objectives, TELBO has a $\gamma$ scaling factor which controls how we scale the $\log p(y|z)$ term in the $\mathrm{elbo}_{\gamma,1}(\mathbf{y}, \boldsymbol{\theta}_y, \boldsymbol{\phi}_y)$ term. We sweep values of $\{1, 50, 100\}$ for this parameter. In general, we find that the effect of this term is smaller on the performance than the $\lambda_y$ term. Based on the setting of this parameter, we find that, for example, the correctness values for fully specified queries change from 82.08 ($\pm$0.56) at $\gamma$=50 to 80.27 ($\pm$0.38) at $\gamma$=1 on validation set for $\mathtt{iid}$ queries.

3. Effect of $\alpha$: We generally find that $\alpha$=1.0 works best for JMVAE across the different choices explored in Wang et al. (2016a), namely, $\{0.01, 0.1, 1.0\}$. For example, decreasing the value of $\alpha$ to 0.1 or 0.01 reduces correctness for fully sepcified queries from 85.63 ($\pm$0.29) to 77.58 ($\pm$0.23) at 0.1 and 74.57 ($\pm$0.44) at 0.01 respectively on the validation set for $\mathtt{iid}$ queries.

4. Effect of $\mu$: For BiVCCA, we ran a search for $\mu$ over $\{0.3, 0.5, 0.7\}$, running each training experiment four times, and picked the best hyperparameter choice across the runs. We found that $\mu$=0.7 was the best value, however the performance difference across different choices was not very large. Intuitively, higher values of $\mu$ should lead to improved performance compared to lower values of $\mu$. This is because lower values of $\mu$ mean that we put more weight on the $\mathrm{elbo}$ term with a $q(\mathbf{z}|\mathbf{x})$ inference network than the one with a $q(\mathbf{z}|\mathbf{y})$ inference network, which results in sharper samples.

## A.7 COMPOSITIONAL GENRALIZATION ON MNIST-A: QUALITATIVE RESULTS AND DETAILS

We next show some examples of compositional generalization on MNIST-A on a validation set of queries. For the compositinal experiments we reused the parameters of the best models on the iid splits for all the models, and trained the models for $\sim 160K$ iterations. All other design choices were the same. Figure 12 shows some qualitative results.

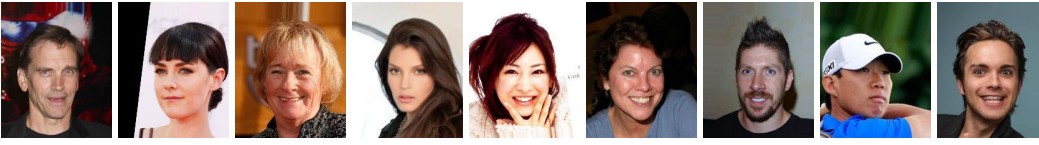

Figure 13: Set of all 9 images labelled as `bald=1` and `male=0` in the CelebA dataset. We can see that in all the cases the labels are inaccurate for the image, probably due to annotator error.

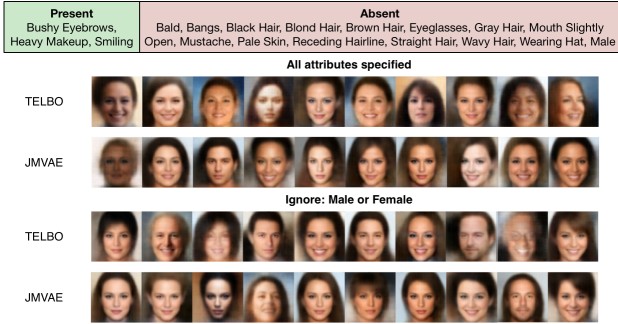

Figure 14: **TELBO creates more diverse images than JMVAE.** At the top we show the set of attributes which are present and absent in the input query. Below, we show the results of generation with all the attributes specified, drawing 10 samples each. We see that both TELBO and JMVAE create accurate images satisfying the constraints. Note that the concept "male" is set to "absent" in the query, which in CelebA means that "female" is present. Next, we unspecify whether the image should contain a male or a female. We see that in this setting, TELBO has a better mixing of male and female images (fourth, sixth, eighth and ninth images in the third row are male), than JMVAE which just produces a single male image (the ninth image in the fourth row).

## A.8 Details on CelebA

CelebA consists of 202,599 face colored images and 40 attribute binary vectors. We use the version of this dataset that was used in (Perarnau et al., 2016); this uses a subset of 18 visually distinctive attributes, and preprocesses each image so they are aligned, cropped, and scaled down to 64 x 64. We use the official train and test partitions, 182K for training and 20K for testing. Note that this is an iid split, so the attribute vectors in the test set all occur in the training set, even though the images and people are unique. In total, the original dataset with 40 attributes specified a set of 96486 unique visual concepts, while our dataset of 18 attributes spans 3690 different visual concepts.

In Section 5.2, we claim that our generations of "Bald" and "Female" images are from a compositionally novel concept. Our claim comes with a minor caveat/clarification: the concept `bald=1` and `male=0` does occur in 9 training examples, but they are all incorrect labelings, as shown in Figure 13! Further, we see that the images generated from our model (shown in Figure 5) are qualitatively very different from any of the images here, showing that the model has not memorized these examples.

## A.9 More results on CelebA

Finally, we show further qualitative examples of performance on the CelebA dataset. We focus on the TELBO and JMVAE objectives here, since BiVCCA generally produces poor samples (see Figure 5). Figure 14 (middle) shows some example generations for the concept specified by the attributes (top). We see that both TELBO and JMVAE produce correct images when provided the full attribute queries (first two rows). However, when we stop specifying attribute "male" or "not male" (female), we see that TELBO provides more diverse samples, spanning both male and female (compared to JMVAE). This ties into the explanation in Appendix A.1, where we show how one can interpret JMVAE as optimizing for the $\mathrm{KL}(q_{\boldsymbol{\phi}}^{\mathrm{avg}}(\mathbf{z}|\mathbf{y}_i)|q_{\boldsymbol{\phi}_y}(\mathbf{z}|\mathbf{y}_i))$ to fit the unimodal inference network $q_{\boldsymbol{\phi}_y}(\mathbf{z}|\mathbf{y}_i)$. Since JMVAE only reasons about the "aggregate" posterior as opposed to the

prior (which TELBO reasons about), it has the tendency to generate less diverse samples when shown unseen concepts.

