# OpenReview forum: "Generative Models of Visually Grounded Imagination"
_ICLR.cc/2018/Conference — Accept (Poster)_

### Official Review · AnonReviewer2 · 2017-11-25
**Clear paper with convincing results**

**Rating:** 7
**Confidence:** 3

**Review:**

The authors propose a generative method that can produce images along a hierarchy of specificity, i.e. both when all relevant attributes are specified, and when some are left undefined, creating a more abstract generation task.

Pros:
+ The results demonstrating the method's ability to generate results for (1) abstract and (2) novel/unseen attribute descriptions, are generally convincing. Both quantitative and qualitative results are provided.
+ The paper is fairly clear.

Cons:
- It is unclear how to judge diversity qualitatively, e.g. in Fig. 4(b).
- Fig. 5 could be more convincing; "bushy eyebrows" is a difficult attribute to judge, and in the abstract generation when that is the only attribute specified, it is not clear how good the results are.

---

> ### Author Response · Authors · 2017-12-20
> **Clarifications**
>
> We are glad the reviewer thought that our paper was clear and has convincing results. Below we provide some clarifications to address the cons raised by the reviewer:
>
> 1. How to judge diversity in Fig. 4 b):
> The new version provides a clearer caption to Fig. 4 b) explaining the qualitative results for MNIST-A. Hopefully this should make judging diversity easier.
>
> 2. Explanation for Fig. 5:
> We have updated the text on Page. 9, to better explain the qualitative results for CelebA, in addition to further examples in Appendix A.9.
>
>   Further, we would like to point out a subtle distinction between attributes which are ‘absent’ and attributes which are ‘unspecified’. ‘Absence’ means that the value of an attribute is set to 0, while unspecified means that we don’t know whether the attribute should be set (1) or unset (0). Thus, for Fig. 5, there are a total of 16 attributes specified (15 unset and 1 set) (for the most abstract generation example, male=*, smiling=*), as opposed to just ‘bushy eyebrows’ being specified (which is one of the premises of the reviewer’s concern).

---

### Official Review · AnonReviewer1 · 2017-11-25
**Well written paper**

**Rating:** 7
**Confidence:** 4

**Review:**

This paper presented a multi-modal extension of variational autoencoder (VAE) for the task "visually grounded imagination."  In this task,  the model learns a joint embedding of the images and the attributes. The proposed model is novel but incremental comparing to existing frameworks.  The author also introduced new evaluation metrics to evaluate the model performance concerning correctness, coverage, and compositionality.

Pros:
1. The paper is well-written, and the contribution (both the model and the evaluation metric) potentially can to be very useful in the community.
2. The discussion comparing the related work/baseline methods is insightful.
3. The proposed model addresses many important problems, such as attribute learning, disentanged representation learning, learning with missing values, and proper evaluation methods.

Cons/questions:
1. The motivation of the model choice of q is not clear.
Comparing to BiVCCA, apart from the differences that the author discussed, a big difference is the choice of q.  BiVCCA uses two inference networks q(z|x) and q(z|y), while the proposed method uses three. q(z|x), q(z|y), and q(z|x,y).  How does such model choice affect the final performance?

2. Baselines are not necessarily sufficient.
The paper compared the vanilla version of BiVCCA but not the one with factorized representation version. In the original VAECCA paper, the extension of using factorized representation (private and shared) improved the performance]. The author should also compare this extension of VAECCA.

3. Some details are not clear.
a) How to set/learn the scaling parameter \lambda_y and \beta_y? If it is set as hyper-parameter, how does the performance change concerning them?
b) Discussion of the experimental results is not sufficient. For example, why JMVAE performs much better than the proposed model when all attributes are given.  What is the conclusion from Figure 4(b)? The JMVAE seems to generate more diverse (better coverage) results which are not consistent with the claims in the related work.  The same applies to figure 5.

---

> ### Author Response · Authors · 2017-12-20
> **Replies to cons/questions**
>
> We are glad the reviewer thought that our contribution would be useful to the community and that our work addresses important problems. Below we address specific cons/ questions raised by the reviewer:
>
> 1. The motivation for the model choice of q:
> In general, when there is information asymmetry in the two modalities (i.e. one instance in modality y corresponds to many instances in modality x), we believe it is beneficial to learn with a joint inference network and retrofit (like TELBO does) as opposed to trying to explain both modalities using the unimodal inference network (say for q(z| y)) where I(y) < I(x), since it is difficult for the latent variable in this case to explain 'all' the instances of the second (information rich) modality.
>
> 2. Comparison to private-VCCA:
> A direct comparison to private-VCCA (from Wang.et.al. (VAECCA)) cannot be made since the private-VCCA model does not have a mechanism to do ‘imagination’. Concretely, the private-VCCA model cannot be applied for multimodal translation, i.e. going from x to y or vice versa, since in this model, p(x| h_x, z) needs access to h_x and z, while the inference networks that condition on 'y' are just q(h_y| y) and q(z| y), meaning there is no path to go from ‘y’ to ‘x’, which is needed for conditional generation.
>   We implemented a modified version of private-VCCA and /extended/ it to the BiVCCA setting (i.e. learning two inference networks) to obtain a version that /can/ do conditional generation. We call this model imagination-private-BiVCCA. Concretely, we learn three inference networks per modality: q(h_x| y), q(h_y|y) and q(z| y). Now, we see that given ‘y’, we can sample h_x and z, and thus, feed it through p(x| h_x, z) to generate images. Similar to Wang. et.al., we regularize the latent space in this model using dropout, searching for values in the range {0, 0.2, 0.4}. As with the BiVCCA model we present in this paper, we also search for \mu in range {0.3, 0.5, 0.7}. We find that this approach does about as well as the best BiVCCA method we already report in the paper and is still substantially worse than JMVAE or TELBO.
>
> 3. Details of the hyperparameters, \lambda_y and \beta_y:
> Firstly, we clarified notation, replacing \beta_y with \gamma in the revised version. We have added details of how performance changes based on different hyperparameters for all the objective functions in appendix A.6. Concretely, the important parameters to tweak for TELBO are \lambda_y (with \gamma) being less important. For JMVAE, both \lambda_y and \alpha turn out to be fairly important, while for BiVCCA, \lambda_y is important while the choice of \mu is marginally important.
>
> 4. Why is JMVAE better at correctness when all attributes are specified?
> Page. 6 explains the tradeoff between correctness and coverage implicit in choices of JMVAE v.s. TELBO objectives. Briefly, our initial submission showed in appendix A.1. that JMVAE fits an inference network optimizing the KL divergence between the aggregate posterior (for the set of images given a label y_i) and the unimodal inference network q(z| y_i), to ‘cover’ the aggregate posterior. This ensures that the samples from q(z| y_i) are better tuned to work with p(x| z), since the elbo(x, y) term has a likelihood function p_{\theta_x}(x| z) which is fed samples from q(z| x, y), whose aggregate posterior q(z| y_i) tries to match. In contrast TELBO regularizes q(z| y_i) to be ‘close’ to p(z) which leads to better coverage, but does not lead naturally to the expectation that we would achieve better correctness than a model like JMVAE, which in some sense has a tighter coupling between the likelihood and the q(z| y) terms. If the aggregate posterior matches the prior better, the gap in correctness between JMVAE and TELBO would reduce. Making this happen is an active research area in variational inference (see [A], [B]).
>
> 5. Clarifications about qualitative results:
> The new version provides a clearer caption to Fig. 4 b) explaining the qualitative results for MNIST-A and adds new text on Page. 9 explaining the qualitative results on CelebA. In addition, we show more qualitative results in Appendix A.9 (Page. 18, Figure. 13) discussing results comparing TELBO and JMVAE in terms of diversity.
>
> References:
> [A]: Tomczak, Jakub M., and Max Welling. 2017. “VAE with a VampPrior.” arXiv [cs.LG]. arXiv. http://arxiv.org/abs/1705.07120.
> [B]: Hoffman, Matthew D., and Matthew J. Johnson. 2016. “Elbo Surgery: Yet Another Way to Carve up the Variational Evidence Lower Bound.” In Workshop in Advances in Approximate Bayesian Inference, NIPS. http://approximateinference.org/accepted/HoffmanJohnson2016.pdf.

---

### Official Review · AnonReviewer3 · 2017-11-27
**nice paper, needs some clarification**

**Rating:** 7
**Confidence:** 3

**Review:**

The paper proposes a method for generating images from attributes. The core idea is to learn a shared latent space for images and attributes with variational auto-encoder using paired samples, and additionally learn individual inference networks from images or attributes to the latent space using unpaired samples. During training the auto-encoder is trained on paired data (image, attribute) whereas during testing one uses the unpaired data to generate an image corresponding to an attribute or vice versa. The authors propose handling missing data using a product of experts where the product is taken over available attributes, and it sharpens the prior distribution. The authors evaluate their method using correctness i.e. if the generated images have the desired attributes, coverage i.e. if the generated images sample unspecified attributes well, and compositionality i.e. if  images can be generated from unseen attributes. Although the proposed method performs slightly poor compared to JMVAE in terms of concreteness when all attributes are provided, it outperforms when some of the attributes are missing (Figure 4a). It also outperforms existing methods in terms of coverage and compositionality.

Major comments:

The paper is well written, and summarizes its contribution succinctly.

I did not fully understand the 'retrofitting' idea. If I understood correctly, the authors first train \theta and \phi and then fix \theta to train \phi_x and \phi_y. If that is true, then is \calL(\theta, \phi, \phi_x, \phi_y) are right cost function since one does not maximize all three ELBO terms when optimizing \theta? Please clarify?

Minor comments:

- 'in order of increasing abstraction', does the order of gender-> smiling or not -> hair color matter? Or, is male, *, blackhair a valid option?

- what are the image sizes for the CelebA dataset

- page 5: double the

- Which multi-label classifier is used to classify images in attributes?

---

> ### Author Response · Authors · 2017-12-20
> **Clarifications on the comments**
>
> We thank the reviewer for the encouraging feedback on the paper. Below we provide clarifications to specific concerns.
>
> 1. Clarifications about “retrofitting”:
> As explained on Page. 3, last paragraph, and by the reviewer, the retrofitting idea is that we first train \theta and \phi and then fix \theta to train \phi_x and \phi_y. As explained in the paper, we just optimize the first term in the cost function with respect to \theta and fix \theta_x and \theta_y when training the last two ELBO terms respectively. This is what we refer to as ‘retrofitting’. To clarify things further, we write the cost in terms of L(\theta_x, \theta_y, \phi_x, \phi_y, \phi) to clarify which parameters get used in each of the ELBO terms. However, conceptually, we still retain the notation specifying the overall cost using \theta for succinctness, and for the possibility of performing semi-supervised learning with the objective (footnote, Page. 4), which is beyond the scope of the current paper.
>
> 2. Order of attributes:
> Thanks a lot for raising this point! We have updated the paper with clarifications (Page. 1) explaining that the order of attributes does not matter.
>
> 3. Image sizes for CelebA:
> The updated version of the paper contains the image sizes used for both MNIST-A and CelebA. The sizes used are the same across both the datasets, namely 64x64.
>
> 4. Details of the multi-label classifier:
> Page. 15 provides more details of the multi-label classifier used for evaluating imagination on MNIST-A dataset. The multi-label classifier is a 2 layer convolutional neural network with four heads, where each head is a 1 hidden layer MLP trained end-to-end to optimize for the log-likelihood of the correct attribute label for an image.

---

### Public Comment · (anonymous) · 2017-12-16
**Reproducibility Assessment**

The challenge that this paper is addressing is to generate images from novel abstract concepts and compositionally novel concrete concepts, input as a set of attribute descriptions. The authors propose a product-of-experts inference network using a joint variational autoencoder model, with a new objective called triple evidence lower bound, or TELBO. It is claimed in the paper that this objective performs on par with same architecture with JMVAE objective and significantly outperforms one with BiVCCA objective in correctness. Also, it is shown in the paper that TELBO outperforms both baselines when the given attributes in an image description are not fully specified (coverage and compositionality).
In an effort to reproduce the results of this paper and confirm the claims, our team worked on a project aiming to replicate the methodology of the authors. In this path, we find the paper well-written and clear. The source code is publicly available on Github, which facilitates the reproduction of the models. While the MNIST-A dataset used in paper is proprietary to Google, the provided datasets are very similar but not identical to the ones that the authors used. Furthermore, the details of the infrastructure and computational setup were not mentioned in the paper.
In our project, we decided to narrow to scope of this project by focusing on reproducing the results of the paper for i.i.d MNIST-A dataset, using the source code provided by the author. We ran our experiments on a Google Compute Engine, using one Nvidia Tesla P100 accelerator. The hyperparameter set that gives the best performance is missing. Our testing results has a significant resemblance with the results in the paper. The average percentage difference between our and their results is at 1.18%. This is a small difference, therefore confirming the methodology and results presented in this paper.
The code implements the architecture and the objective functions described in the paper. Also, our quantitative results match nicely with those in the paper. However, we noticed a misalignment in the qualitative results, namely between the reproduced image and the images in the paper. Our reproduced images are not found in the paper, creating some confusion. Therefore, we were not able to obtain a full understanding the process that transforms the raw images we have to the findings presented in the paper. We believe that with more clarification, we can do more qualitative analysis on the output images.
A key difference between our experiments and those of the authors is the number of steps that we used. We ran TELBO and JMVAE for 250,000 steps, and BiVCCA for 50,000 steps, while the authors trained their models for 500,000 steps. The similarity in coverage and correctness implies that 250,000 steps is enough to train a well-performed model. The additional steps in author’s experiments did not lead to overfitting, but it did not make a significant improvement either. Moreover, the author noted in their paper that the models do not tend to overfit in their experiment. Indeed, our reproduced results verify this claim.
Overall, the process of reproducing this paper’s result is straightforward. Feasibility of the experiments largely depends on the time and hardware setup. Given more time and resources, we believe that we could reproduce additional experiments such as examining different hyperparameter settings and running the models on CelebA dataset.

You can find our paper at this address: https://goo.gl/kscwAA

---

### Public Comment · (anonymous) · 2017-12-16
**Reproducibility of this paper**

The paper focuses on the ability to create diverse images corresponding to novel concrete or partially specified (abstract) concepts, on both familiar and unseen concepts. The authors define a set of simple evaluation metrics for assessing a thorough understanding of concepts by the generative model. The evaluation metrics introduced by the authors present a new way of evaluating the deep understanding of concepts for generative models in the case of abstract or unseen queries. The learning is carried out through joint embedding of images and attributes. The proposed method presents a new objective function that takes into account two additional unpaired inference networks. The authors have also proposed a methodology to handle missing data, through product of experts. The results show that the proposed method performs better than all other methods on abstract queries and the baseline JMVAE method performs better than other methods on unseen queries. The experiments are conducted on different joint multimodal variational auto-encoder models differentiated by their objective functions, namely BiVCCA, JMVAE and TELBO.

Having access to the entirety of the code, we have reproduced one of the baseline methods presented in the paper,  the JMVAE method, a method proposed by (Suzuki et al., 2017), against which the proposed method is compared. This method has been reproduced against a fairly similar version of the MNIST-A dataset used in the paper both on a iid and compositional split. The error on the compositional split was ~1%. The error on the coverage ranges from ~ 0.5 to 4%. The error on the correctness ranges from ~ 2 to 11%. However, the model has been trained on far fewer steps than those trained by the authors (20,000 against 183,564 for the compositional split and 62,862 against 266,054 for the iid split), access to time, expertise and computational power was limited. The limitations on time were 3 weeks, and the computational power available was a simulated environment on a Google Compute Engine with a NVIDIA Tesla K80 GPU. We did not have access to the Slurm Workload Manager used by the authors of the paper. The generated samples presented in the paper were in general different from the images resulting from our experiments, which was expected, as the authors mention the examples being cherry-picked. The paper has presented a detailed structure of the model, which closely align with the source code. All the result are clearly represented for the three different models.

In conclusion, the requirements for reproducing this paper were clear and concise throughout the process. As the models are expensive and complex, the main limitations were time and computational power. We are confident that  the entirety of the paper could have been reproduced had it been more time and a more powerful hardware structure.

---

### Author Response · Authors · 2017-12-20
**Update to the paper addressing concerns**

We thank the reviewers for the thoughtful feedback and are encouraged the reviewers thought the paper was well written and clear (R1, R2, R3), addresses many important problems (R1), and makes useful contributions (R1).

We also thank the community for providing public comments on reproducibility of our work. We are glad that our conclusions were found to be reproducible and well substantiated, and will incorporate specific feedback for improving reproducibility.

We address the major points raised by the reviewers in an updated version of the paper. The key text to look at is highlighted in blue. Below we provide a summary of the major changes.

1. (R1, R2) Better explanation of qualitative results for both MNIST-A and CelebA:
The new version provides a clearer caption to Fig. 4 b) explaining the qualitative results for MNIST-A and adds new text on Page. 9 explaining the qualitative results on CelebA. In addition, we show more qualitative results in Appendix A.9 (Page. 19, Figure. 13) comparing TELBO and JMVAE in terms of diversity.

2. (R1) Explanation of hyperparameter choices:
Thanks for raising this point. The new version adds a discussion of how the hyperparameter choices for different objectives change performance in Appendix A.6.

3. (R1) Clarification on why JMVAE generates more correct samples:
Page. 6 explains the tradeoff between correctness and coverage implicit in choices of JMVAE v.s. TELBO objectives. In particular, while the earlier version explained why TELBO has better coverage, this version also adds the explanation for why JMVAE has better correctness for seen concepts.

4. (R3) Clarification on the order of attributes being important v.s. Not:
Page. 1 adds text to clarify that the order in which the attributes are dropped is not important and that the tree shown in Fig. 1 is just a subset of the overall DAG.

5. (R3) Details of the multi-label classifier used for evaluation:
Page. 15 provides more details of the multi-label classifier used for evaluating imagination on MNIST-A dataset.

6. We add more clarifications explaining how we pick hyperparameters (Page. 4, Page.7), typical training times for our models (Page. 7), and provide the size of the images we generate (Page. 7) (R3).

As mentioned earlier, all the changes above can be conveniently located by following text highlighted in blue across the paper. We elaborate further on these points and address specific reviewer concerns in individual replies.

---

### Decision · Program_Chairs · 2018-01-29
**ICLR 2018 Conference Acceptance Decision**

**Decision:**

Accept (Poster)

**Comment:**

All three reviewers recommend acceptance. Good work, accept